

# Optimal estimation of cloud properties from thermal infrared observations with a combination of deep learning and radiative transfer simulation

He Huang[1], Quan Wang[1], Chao Liu[2], Chen Zhou[1]

[1] School of Atmospheric Sciences, Nanjing University, Nanjing, 210023, China

[2] Collaborative Innovation Center on Forecast and Evaluation of Meteorological Disasters, Nanjing University of Information Science and Technology, Nanjing, 210044, China

*Correspondence to*: Chen Zhou (czhou17@nju.edu.cn)

**Abstract.** While traditional thermal infrared retrieval algorithms based on radiative transfer models (RTM) could not effectively retrieve the cloud optical thickness of thick clouds, machine learning based algorithms were found to be able to provide reasonable estimations for both daytime and nighttime. Nevertheless, stand-alone machine learning algorithms are occasionally criticized for the lack of explicit physical processes. In this study, RTM simulations and a machine learning algorithm are synergistically utilized using the optimal estimation (OE) method to retrieve cloud properties from thermal infrared radiometry measured by Moderate Resolution Imaging Spectroradiometer (MODIS). In the new algorithm, retrievals from a machine learning algorithm are used to provide priori states for the iterative process of OE method, and an RTM is used to create radiance lookup tables that are used in the iteration processes. Compared with stand-alone OE, the cloud properties retrieved by the new algorithm show an overall better performance by using statistic priori information obtained by machine learning algorithm. Compared with stand-alone machine-learning based algorithm, the radiances simulated based on retrievals from the new method align more closely with observations, and physical radiative processes are handled explicitly in the new algorithm. Therefore, the new method combines the advantages of RTM-based cloud retrieval methods and machine-learning models. These findings highlight the potential for machine-learning-based algorithms to enhance the efficacy of conventional remote sensing techniques.

## 1 Introduction

Clouds play an important role in the Earth's energy budget by altering radiation patterns at both the surface and the top of the atmosphere (TOA) (Liou and Davies, 1993; Stubenrauch et al., 2006). Cloud





properties change in response to variations in greenhouse gases, aerosol concentrations, and global

surface temperature, leading to large uncertainties in climate change projections (Forster et al., 2021; Sassen et al., 2007). Grasping the variations in cloud properties is crucial for a comprehensive understanding of cloud dynamics and their radiative impacts on global climate change. The advancement of science and technology has positioned satellite remote sensing as a pivotal tool for monitoring cloud behaviors across diverse spatial and temporal scales. Active satellites like CloudSat and CALIPSO

(Cloud-Aerosol Lidar and Infrared Pathfinder Satellite Observations) offer unparalleled cloud profiling capabilities (Marchand et al., 2008; Sassen et al., 2009). Conversely passive satellites, renowned for their extensive swath observations, are widely applied in a range of atmospheric research.

In recent decades, numerous efforts have been made to retrieve cloud properties using passive satellite instruments (Lai et al., 2019; Li et al.,2023; Min et al., 2020; Minnis et al., 2011; Poulsen et al.,

2012; Tan et al., 2022; Zhao et al., 2012). A common method involves combining data from visible (VIS) and near-infrared (NIR) channels to construct lookup tables (LUT) for daytime cloud microphysical properties, such as cloud optical thickness (COT) and cloud effective radius (CER) (Painemal and Zuidema, 2011; Twomey et al., 1980; Nakajima et al., 1990). This approach is grounded in the principle that cloud reflectance in non-absorbing VIS wavelengths predominantly signifies COT, while reflectance

in absorbing NIR wavelengths is closely related to cloud effective radius (Arking and Childs, 1985; Rossow et al., 1989). Additionally, distinguishing liquid water from ice clouds using NIR channels (e.g., 1.65 μm) has also proven beneficial for deriving cloud top height (CTH) (Harshvardhan et al., 2009; Menzel et al., 2008; Pilewskie et al., 1987). Nonetheless, these VIS/NIR-based methodologies are confined to daytime operations owing to their reliance on incident solar radiation, absent during nighttime

hours.

Alternatively, night-time cloud properties can be retrieved using thermal infrared (TIR) radiometry from passive satellite. Inoue (1985) employed the split-window method, leveraging brightness temperature (BT) and BT differences across various window channels, to derive COT and CER. Subsequently, numerous improvements and enhancements have been made to this method (Hamada and

Nishi, 2010; IWABUCHI et al., 2018; Yang et al., 2005). Wang et al. (2016a) implemented an optimal estimation-based (OE) algorithm with Moderate Resolution Imaging Spectroradiometer (MODIS) infrared (IR) observations for cloud property retrieval, demonstrating the suitability of IR channels for



thin ice cloud properties during both daytime and nighttime (Wang et al., 2016b). In addition, the $CO_2$-slicing method, which utilizes adjacent ~15 μm $CO_2$ absorption channels, is able to retrieve CTH

effectively (Smith et al., 1974; Menzel et al., 1983). The atmospheric window IR measurements, such as at 11 μm, are also useful for CTH determination by comparing with the ambient atmospheric temperature profile (Garrett et al., 2009; Hong et al., 2007). However, IR window methods are less effective for optically thick clouds as their BT nears asymptotic values (Garrett et al., 2009; Iwabuchi et al., 2016). While far infrared channels are useful for ice clouds with substantial optical thickness (Libois et al.,

2017), their limited presence on most current satellites limits their application. Moreover, the retrieval methods based on plane-parallel cloud radiative transfer (RT) models face global application challenges due to their high computational demands (Wang et al., 2013).

       Recently, machine learning techniques such as random forests, artificial neural network, and deep learning have gained significant traction in remote sensing (Bai et al., 2021; Guo et al. 2022; Shi et

al.,2020; Tan et al.,2023; Yuan et al., 2020; Zhao et al., 2023). Häkansson et al. (2018) used a neural network algorithm to retrieve cloud top properties from several passive polar orbit sensors, greatly improving CTH retrievals. Advanced machine learning algorithms have particularly enhanced CTH retrievals for high and thin clouds (Min et al., 2020). Wang et al. (2022) developed a convolutional neural network (CNN)-based framework (TIR-CNN), utilizing TIR radiometry from MODIS to retrieve COT,

CER, and CTH. This method demonstrates satisfactory performance compared to both passive and active cloud products and is effective during both daytime and nighttime(Wang et al., 2022, 2023). Tana et al. (2023) obtained cloud detection and cloud microphysical properties with high spatial-temporal resolutions from TIR spectral channels of Himawari-8 using a machine learning algorithm. Zhao et al. (2023) applied a deep-learning ResUnet model for retrieving cloud phase (CLP), COT, CER, and CTH

using FY4A satellite observations.

       However, the reliance of these machine learning methods on mathematical and statistical approaches typically leads to an implicit assimilation of the relationships between cloud properties and radiance observations, lacking direct physical interpretation. A great number of cloud property users favor remote sensing products that offer explicit physical interpretations. Therefore, enhancing

traditional inversion algorithms with machine learning algorithms can be beneficial.



In this study, we integrate traditional radiative transfer simulations with TIR-CNN retrievals using the OE method (OE-CNN-IR) to retrieve COT, CER, and CTH from MODIS, which is effective under both daytime and nighttime conditions. The Community Radiative Transfer Model (CRTM) is utilized to simulate MODIS IR observations and generate LUT for cloud properties. The TIR-CNN retrievals are

employed as priori states, and an iterative process based on gradient descent is performed to get an optimal estimation. The performance of the proposed OE-CNN-IR model is subsequently compared with a stand-alone OE method utilizing fixed priori states. Details of the data and the enhanced OE-CNN-IR method are presented in Section 2. Section 3 outlines the retrieval results and their evaluation against cloud products from passive and active sensors. Conclusions are summarized in Section 4.

**2 Data and Methodology**

**2.1 Data**

**2.1.1 MODIS data**

This study utilizes global data observed by MODIS instrument on the Aqua spacecraft. Aqua-MODIS continuously monitors the earth-atmosphere system with 36 spectral bands ranging from 0.405 to 14.385

μm. For this research, the Aqua-MODIS official Collection 6.1 (C6.1) products (MYD021KM, MYD03, MYD35, MYD06 and MCD12C1), available at https://ladsweb.modaps.eosdis.nasa.gov/search/, have been selected for this study. These products, with spatial resolution of 1km and 5km, are chosen for their widely accepted quality (Wang and Christopher, 2003). In this study, the TIR radiations from Aqua-MODIS collection 6.1 (C6.1) Level 1B calibrated radiances products (MYD021KM) are converted to

BTs using the Planck function. Additionally, atmospheric parameters from MYD03 and MYD06, including surface temperature, land surface type, and cloud phase, are used as ancillary data for LUT construction and forward radiative simulations. Cloud optical and physical parameters such as COT, CER and CTH from MYD06 serve to verify the accuracy of daytime retrievals. All parameters are aligned to a 5-kilometer spatial resolution grid, ensuring data and variable consistency. To validate the universality

of the inversion algorithm, retrievals are conducted using data from one representative day each month in 2009, capturing the variability of atmospheric conditions throughout the year and facilitating a comprehensive evaluation across different scenarios. The retrievals are performed over the whole globe, and the data between 60°S-60°N are used in the validation. By selecting days representative of each





month, we aim to assess the algorithm's performance under varying seasonal and weather patterns. Table

1 summarizes the data and parameters used in our retrieval model.

**Table 1. Summary of MODIS data sources and preprocessing parameters.**

| Product name | Spatial Resolution | Variables | Unit |
|---|---|---|---|
| MYD021KM | 1 km | Band 27 | W/ (m² µm sr) |
| | | Band 28 | W/ (m² µm sr) |
| | | Band 29 | W/ (m² µm sr) |
| | | Band 31 | W/ (m² µm sr) |
| | | Band 32 | W/ (m² µm sr) |
| | | Band 33 | W/ (m² µm sr) |
| | | Band 34 | W/ (m² µm sr) |
| | | Band 35 | W/ (m² µm sr) |
| | | Band 36 | W/ (m² µm sr) |
| MYD03 | 1 km | Sensor Zenith | ° |
| | | Land/Sea Mask | - |
| MYD06 | 1km | Cloud Effective Radius | µm |
| | | Cloud Optical Thickness | - |
| | | Cloud water path | Kg/m² |
| | | Cloud Phase Optical Properties | - |
| | 5km | Cloud Phase Infrared | - |
| | | Cloud Top Pressure | hPa |
| | | Surface Temperature | K |

**2.1.2 Active Lidar Detection cloud products**

Cloud-Aerosol Lidar with Orthogonal Polarization (CALIOP), a space-based lidar instrument onboard

the Cloud-Aerosol Lidar and Infrared Pathfinder Satellite Observations (CALIPSO) satellite, provides

vertical profiles of clouds and aerosols in Earth's atmosphere. CALIOP can perform observations at both

daytime and nighttime, overcoming the limitations of passive optical instruments, but it could not



penetrate thick clouds. The Cloud Profiling Radar (CPR) aboard the CloudSat satellite is a radar system that sends out microwave pulses and measures the reflected energy from clouds. This technique is particularly adept at determining the structure and ice content within clouds, but fails to detect thin clouds.

The DARDAR product (Delanoë and Hogan, 2010), integrating data from both CALIOP and CPR, offers a comprehensive atmospheric column view that neither instrument can achieve independently. This extensive dataset includes information on cloud top and base heights, optical thickness, ice content, and aerosol layers. In our study, the ice cloud product of DARDAR is used to evaluate the inversion results during both daytime and nighttime conditions.

**2.2 Development of the retrieval algorithm**

Figure 1 illustrates the architecture of our retrieval models. Initially, atmospheric parameters including temperature, humidity and ozone from the Fifth Generation of the European Centre for Medium-Range Weather Forecasts (ECMWF) Reanalysis (ERA5) are used to construct lookup tables for each 0.25°x0.25° spatial grid box. These LUTs enumerate the BT for each channel corresponding to varying COT, CTH

and CER. Subsequently, the OE method is performed to retrieve cloud properties. In this approach, the TIR-CNN derived cloud properties provide priori states for iterative processes, which is subsequently refined through iterative minimization of the objective cost function. This method iteratively adjusts parameters to reconcile observed data with model predictions. Further details are presented below.

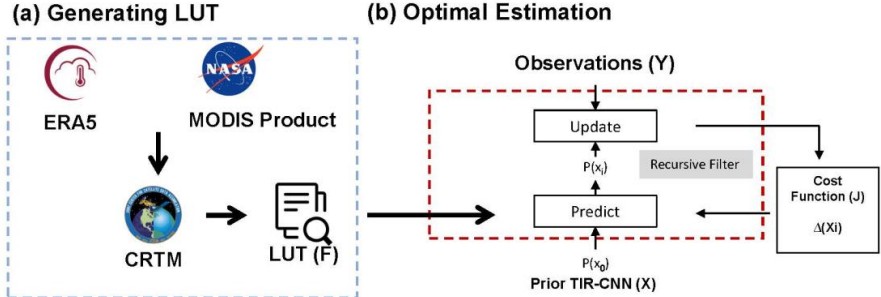

**Figure 1. The architecture of the retrieval model. (a) The establishment of look-up table. (b) The iteration steps in the optimal estimation progress.**

**2.2.1 Forward Model**

The CRTM, developed by the U.S. Joint Center for Satellite Data Assimilation (JCSDA), spans a broad spectrum of channels from visible to microwave. It is widely used in simulating radiances at the top of

the atmosphere for various satellite sensors, owing to its flexible interface, sophisticated radiative transfer processes, and efficient numerical computation(Han et al., 2006). The model divides the atmosphere into a series of vertical layers, and the temperature, pressure and composition of each layer is assumed to be homogenous. CRTM solves the radiative transfer equations throughout the atmosphere, and is able to simulate the radiances observed by satellites. Its precision and reliability have been extensively

corroborated by ground-based and satellite observations(Zou et al., 2016). Considering that the optical properties of ice cloud crystals in CRTM and MODIS product is different due to differences in particle habit assumptions (Yi et al., 2016; Yao et al., 2018), the volumetric extinction cross section in CRTM is adjusted by a scaling factor (0.4), and the simulated brightness temperature is consistent with observations (Fig. 5(a,b,c)). For each grid cell, the CRTM simulates TIR radiances corresponding to

various COT, CER, and CTH values at each location, from which a LUT is subsequently constructed. Table 2 provides a detailed list of the cloud properties and ancillary parameters used in these calculations.

**Table 2. Geometries and cloud properties selected to calculate the cloud lookup tables**

| | Variable Names | Notes |
|---|---|---|
| Reference cloud properties | COT | 0.01, 0.03, 0.05, 0.10, 0.20, 0.30, 0.40, 0.50, 0.60, 0.70, 0.80, 0.90, 1.00, 1.20, 1.40, 1.60, 1.80, 2.00, 2.50, 3.00, 3.50, 4.00, 4.50, 5.00, 5.50, 6.00, 6.50, 7.00, 7.50, 8.00, 8.50, 9.00, 9.50, 10.0, 12.0, 15.0, 20.0, 25.0, 30.0, 50.0 |
| | CER(μm) | 5, 10, 15, 20, 25, 30, 35, 40, 45, 50, 55, 60, 65, 70, 75, 80, 85, 90 |
| | CTH(km) | 0.1,0.8,1.15,1.5,2,2.5,3.5,5,6.25,8,10,12,14,16 |
| Model parameter | Surface temperature(K) | MYD06 |
| | Land type | MCD12C1, IGBP |
| | Cloud type | MYD06, Cloud Phase |
| | Temperature profile(K) | ERA5, Temperature |



| Water vapor profile(g/kg) | ERA5, Specific humidity |
|---|---|
| Ozone profile(g/kg) | ERA5, Ozone mass mixing ratio |

The outputs of the forward model can be expressed as a function of cloud properties and ancillary parameters:

$$Y = [BT_1 BT, BT_2, \ldots \ldots, BT_m]^T = F[X(COT, CER, CTH), P] + e, \tag{1}$$

where Y is a vector consisting of m MODIS IR observations in BT, P is a vector encompassing various ancillary variables, including air temperature, water vapor concentration, ozone concentration profiles, surface emissivity spectrum, and surface temperature, and e is an error term.

Figure 2 depicts the variation in CRTM output (F) (expressed in BT) as a function of ice cloud properties, derived from a simulation using the atmospheric profile dated June 10, 2009, at 00:00 UTC, at coordinates 175.87°E longitude and 60.55°N latitude. With fixed CER and CTH, the TOA BTs in MODIS IR bands generally decrease with increasing COT. Notably, for COT>10, the slopes approach zero, causing challenges in inversion accuracy. In the case of fixed COT and CTH, TOA BTs decrease with increasing CER values when CER is below 10 μm across all channels, followed by minor oscillations in most channels, except that band 29 shows significant variations. For CTH values under 11 km, TOA BTs are negatively correlated with CTH noticeably.

Figure 3 shows the relationship between TOA BTs and liquid cloud properties, which reveals a weaker response to changes in COT and CER compared to ice clouds. Nevertheless, TOA BTs decrease noticeably with increasing water cloud CTH. In summary, CTH is the most accurately determinable variable for both ice and water clouds due to the high sensitivity of TOA BTs to CTH. For ice clouds, COT values below 10 generally allow for more accurate retrieval of cloud properties in theory. However, retrieving CER for ice clouds poses greater challenges due to the complexity of ice particle size distribution and shape. Water clouds, conversely, show no strong sensitivity of TOA BTs to both COT and CER, and it is more difficult to accurately retrieve these cloud properties solely based on TOA BT observations.

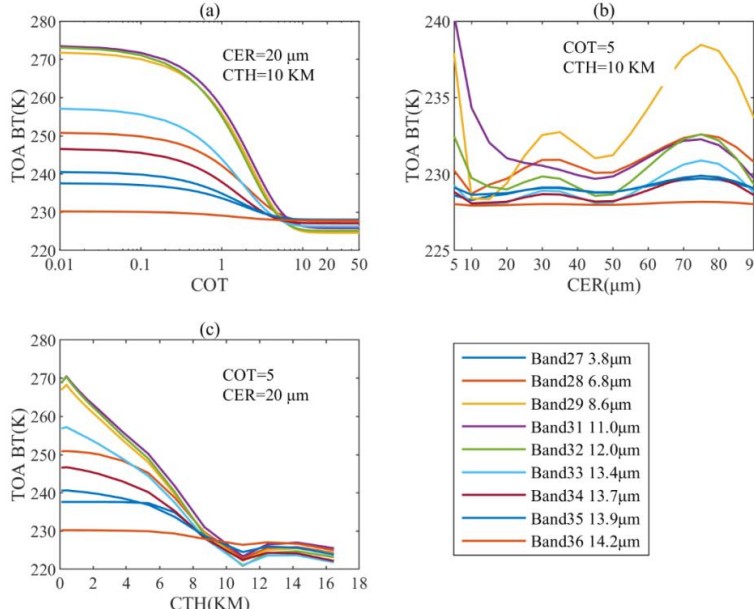

**Figure 2. Radiative transfer model simulations for ice clouds. The atmospheric profile is from the coordinates with a longitude of 175.87°E and a latitude of 60.55°N, on June 10, 2009, at 00:00 UTC. (a) TOA BTs as a function of COT, when CER and CTH is set to 20 μm and 10 km, respectively. (b) BT as a function of CER, when COT and CTH is set to 5 and 10km, respectively. (c) BT as a function of CTH, when COT and CER is set to 5 and 20 μm, respectively.**


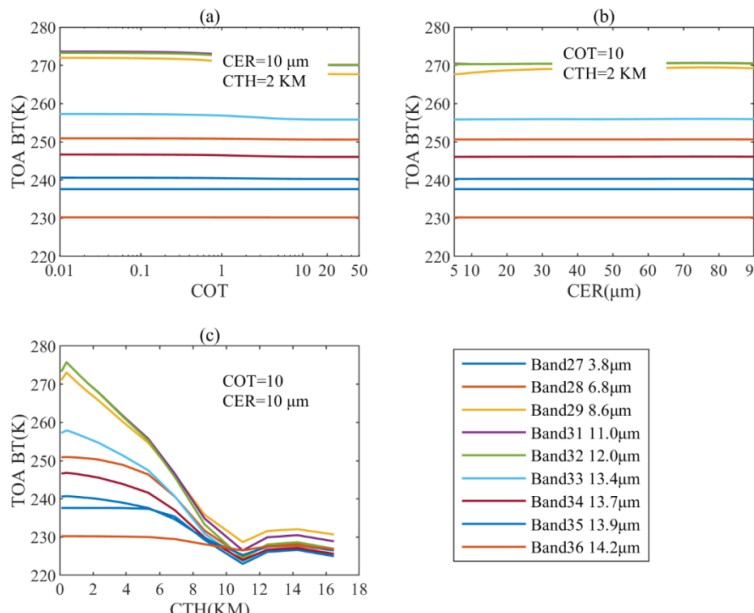

**Figure 3. Same as Fig. 2, but for liquid clouds.**

### 2.2.2 The Convolutional Neural Network Infrared method


The convolutional neural network using thermal infrared (TIR-CNN) model is trained with solar-independent variables (thermal infrared radiances, viewing zenith angles, and altitude) as inputs and uses standard MYD06 products (solar-dependent retrievals in the daytime) as targets. Through training, the model can capture context and learn the complex nonlinear relationship between the input variables and

targets, which can be applied in the cloud property retrievals during both daytime and nighttime. The convolutions in the TIR-CNN model are beneficial in considering statistic information from neighbor fields in training. Theoretically, spatial distributions, optical and microphysical properties of clouds are all determined by the meteorological backgrounds, so cloud properties are statistically connected to their horizontal distributions. In addition, the effective radius of ice cloud particles are functions of cloud

temperature. The CNN-based deep learning architecture is able to capture the statistical features among adjacent pixels of satellite observations as a solution for retrieving cloud optical and micro-physical properties (Wang et al., 2022, 2023), so it is able to provide more information than traditional algorithms that retrieve cloud properties from infrared radiances of individual pixels. The benefits of machine





learning in IR cloud retrievals have also been demonstrated independently by the results of Tana et al.

(2023) and Zhao et al. (2023).

### 2.2.3 Optimal Estimation-Based Retrieval Method

The OE-based retrieval method, as introduced by (Rodgers, 2000), is designed to derive the best estimates

of atmospheric quantities (such as temperature, humidity, aerosol concentration, or trace gas

concentrations) by minimizing the discrepancy between observed measurements and the model

predictions. This method combines information from both the measurement data and a priori knowledge,

typically obtained from atmospheric models or ancillary data sources. A key strength of the OE method

is its proficiency in addressing complex atmospheric retrieval challenges, enabling simultaneous retrieval

of multiple parameters in contexts where physical processes are nonlinear and highly coupled. It provides

a rigorous and statistically robust method to estimate atmospheric parameters, along with quantifying the

associated uncertainties.

The OE method aims to identify the most probable state variables by minimizing a cost function J:

$$J = [F(X,P) - Y]^T S_y^{-1} [F(X,P) - Y] + [X - Xa]^T S_a^{-1} [X - X_a], \tag{2}$$

where $X_a$ and $X$ are the prior and posterior state vectors, respectively. $S_y$ and $S_a$ are the covariance

matrices of the observation-to-simulation differences and the uncertainty of the prior state vector,

respectively. Then we employ an iterative process to find an optimal solution based on observed data and

priori states. Mathematically, the gradient descent iterative process for *(i+1)*'th iteration is encapsulated

by:

$$X_{i+1} = X_i - \theta \frac{\partial J_{i,n}}{\partial X_{i,n}} , \tag{3}$$

where

$$\frac{\partial J_{i,n}}{\partial X_{i,n}} = \frac{J(X_{i,n} + \delta x_n) - J(X_{i,n})}{\delta x_n} , \tag{4}$$

and $\theta$ represents a learning rate and n represents the n-th cloud parameters (COT, CER and CTH), and

is set to be the same for all three variables. In this paper, $\theta$ is initially set to 0.05 for the first 200 iterations

and after those initial 200 iterations, the learning rate is then reduced to 0.01 for the subsequent 100

iterations. $\delta x_n$ represent the small increase in n cloud parameters and $J(X_{i,n} + \delta x_n)$ are calculated

using LUTs.

In this paper, stand-alone OE-IR method relies on a fixed priori value for its iterative process, whereas OE-CNN-IR utilizes results from TIR-CNN as its priori states for further refinement. These methods illustrate the integration of traditional optical estimation techniques with advanced machine learning models to potentially enhance the accuracy and reliability of atmospheric measurements.

Figure 4 shows the iterative variations in cost function, COT, CER and CTH for both OE-IR and OE-CNN-IR under various conditions. For smaller cot values (COT < 10), OE-CNN-IR and OE-IR exhibit consistent effects on COT and CTH, converging with close values. For both methods, the CER depends on priori states. A distinct difference is that OE-CNN-IR starts with a significantly lower cost than OE-IR and maintains more stability throughout the iteration process. For larger COT values (COT > 10), the

CTH of these two methods converge to same value, despite of differences in priori states. The COT of OE-IR struggles to iterate towards the expected target during the iteration process, while OE-CNN-IR maintains stably around the priori values. The iterative results indicate that both methods perform well on COT and CTH for small COT values. However, for large COT values, the OE-IR method fails to function effectively. In contrast, the OE-CNN-IR is able to retrieve COT of thick clouds effectively.

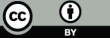

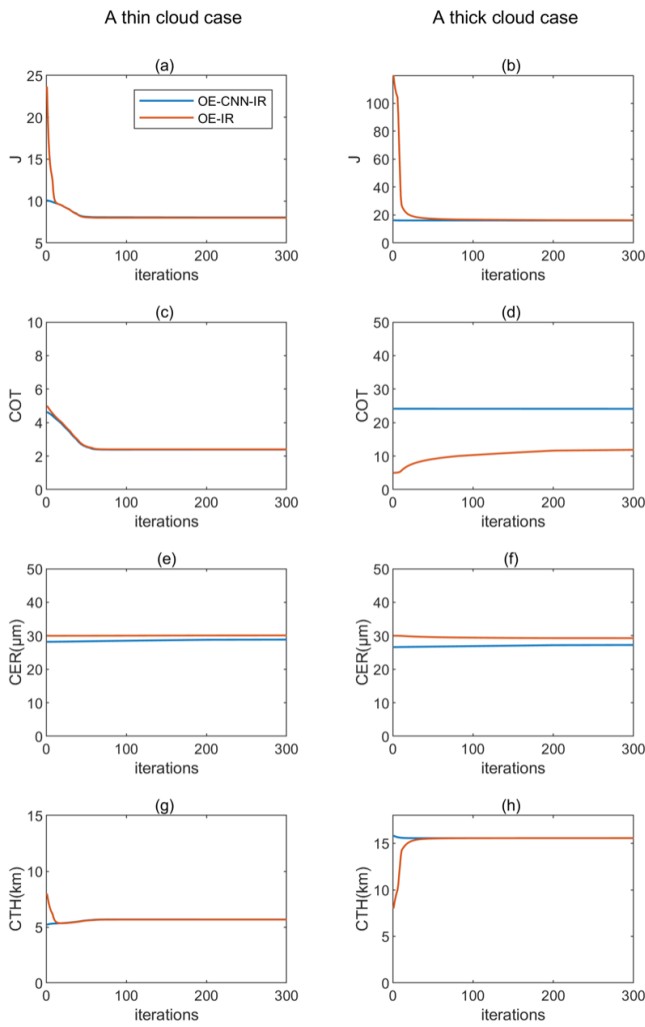

**Figure 4. The change of cost function and cloud parameters in the iteration processes, with OE-IR in Red and OE-CNN-IR in Blue. Left pictures are for an illustrative ice cloud layer with a small optical thickness case and right pictures are for a large optical thickness case.**

**2.3 Metrics for performance evaluation**

In this study, the magnitude of forecast errors, systematic bias errors, and linear correlation between outputs and standard values are quantitatively assessed using three key statistical metrics: root mean squared error (RMSE), mean bias error (MBE), and the Pearson correlation coefficient (r).

$$RMSE = \sqrt{\frac{\sum_{i=1}^{N}(y_i - f_i)^2}{N}} \ , \tag{5}$$



$$MBE = \frac{\sum_{i=1}^{N} f_i - y_i}{N} \,, \tag{6}$$

$$r = \frac{\sum_{i=1}^{N}(f_i - \bar{f})(y_i - \bar{y})}{\sqrt{(\sum_{i=1}^{N}(f_i - \bar{f})^2)}\sqrt{(\sum_{i=1}^{N}(y_i - \bar{y})^2)}} \,, \tag{7}$$

where N is the total number of calculated points, y and f are the true and estimated values, respectively.

## 3 Results and discussion

### 3.1 Case studies of OE-CNN-IR and OE-IR retrievals

To illustrate the daytime efficacy of the proposed method, a granule from Aqua-MODIS, captured at 03:00 UTC on June 10, 2009, has been chosen. This particular granule spans the southwestern Pacific Ocean, encompassing the geographical region from 0° to 20°S latitude and from 150°E to 175°E longitude, as depicted in Fig. 6. Figure 5 compares observed BT with those derived from CRTM. Figures 5(a-c) show CRTM-simulated radiances using baseline MODIS cloud products, serving as a control scenario for comparative analysis. The correlation coefficients for channels 29, 31, and 32 are 0.877, 0.905 and 0.891, respectively, indicating CRTM's proficiency in simulating MODIS cloud products. However, there is a persistent negative MBE across these channels. Figures 5(d-f) present a comparison between observations and BT simulated by CRTM and TIR-CNN retrievals, with outcomes that are analogous to those depicted in Figs. 5(a-c). The pronounced correlation indicates that CNN-based inputs proficiently replicate the spatial and radiometric features of clouds, showing high consistency with MODIS MYD06 products. When OE-CNN-IR or OE-IR cloud property retrievals are used to simulate BT, the correlation coefficients between the simulated BT and observations increases significantly, and the absolute values of MBE and RMSE decreases significantly (as shown in Figs. 5g-l). The improvement is attributed to the OE iterations, which reduce the discrepancy between simulated and observed BT. The results indicate that retrievals of the OE-CNN-IR methods align more closely with BT observations compared to the stand-alone TIR-CNN model.

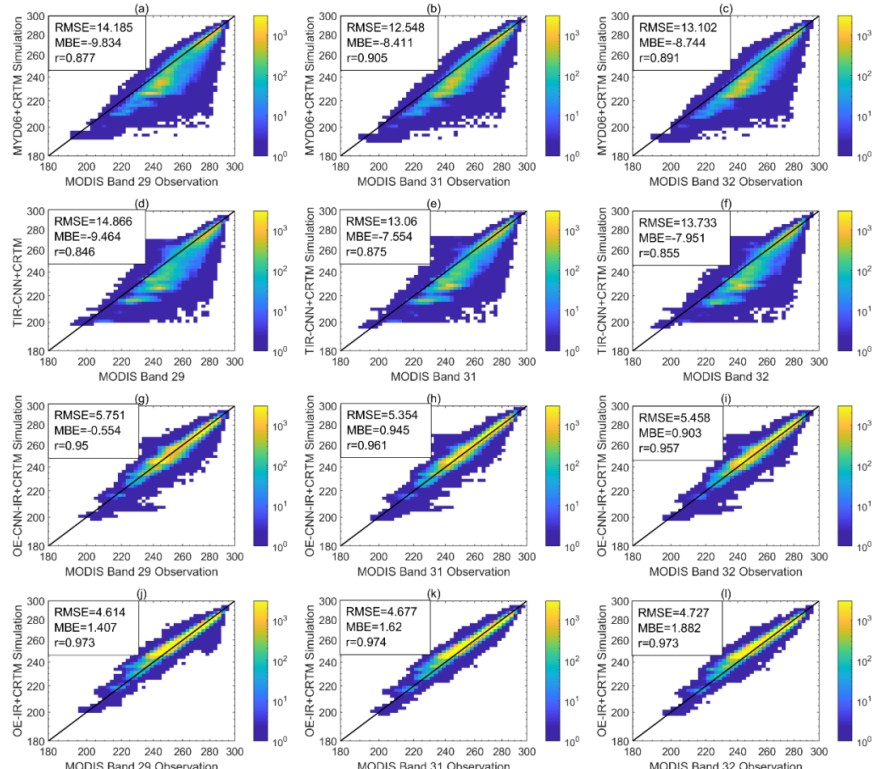

**Figure 5. Comparison between MODIS BT observations and simulated BT based on MODIS cloud properties (upper), TIR-CNN priori inputs (middle), OE-CNN-IR estimations (lower) and OE-IR estimations(bottom), based on an illustrative granule of 10 June 2009 (Fig. 6). The first column is the comparison between the simulation and observation of band 29, the middle column is for band 31, and the right column is for band 32.**

Figures 6 (a-c) show the spatial distribution of BT for each respective channel. These measurements reveal variations in thermal radiation, which correlated with cloud properties specific to the wavelengths of the channels used. Figures 6(d-f) show the cloud physical properties as derived using standard MODIS retrieval algorithms. The COT, CER, and CTH from the MYD06 product provide a benchmark for comparison with other inversion methods. The analysis of BT from channels 29, 31, and 32 shows a clear negative correlation with both COT and CTH, and regions with higher BT typically correspond to clouds with smaller optical thickness and lower cloud top heights. This is in line with the principle that thinner clouds permit more infrared radiation to escape from the Earth's surface and atmosphere, leading to higher observed BT. Furthermore, the analysis indicates that clouds with higher BT generally have lower



altitudes. The patterns in Fig. 6, which display cloud properties derived from various inversion techniques, corroborate the physical relationships illustrated in Figs. 2 and 3. These results support the hypothesis that BT can serve as effective proxies for key cloud properties like COT and CTH, essential for comprehending cloud dynamics and their effects on weather and climate systems. Figures 6 (g-i) present

the retrieval results from the deep learning algorithm TIR-CNN method. The CNN-derived retrievals are not only consistent with MYD06 products in spatial patterns, but also agree well with the magnitudes of results. Figures 6 (j-l) present the retrieval results from the OE-CNN-IR method, showing similar spatial distributions to the standard MYD06 products for COT and CTH. However, significant differences are noted in CER retrieved by OE-CNN-IR and MYD06 products, which is consistent with Wang et al.

(2016), which highlighted substantial discrepancies in CER retrieved using OE-IR and VNIR/SWIR/MWIR methods.

    Figures 6 (m-o) show the results retrieved using the traditional OE-IR method with climatological priori states, which employs climatological values of COT, CER and CTH as priori states for OE iteration. In case where COT values are below a certain threshold, the OE-IR COT closely match MYD06 products,

indicating that it is able to capture the COT of thinner clouds. However, the inability of OE-IR to retrieve COT values greater than 10 suggests a limitation in the technique's sensitivity to optically thicker clouds, aligning with findings from Wang et al. (2016). This threshold effect arises from the TIR BT independence to COT in thick clouds (as shown in Fig. 2a). The performance of CER and CTH retrievals using the OE-IR method is comparable to that of the OE-CNN-IR method.

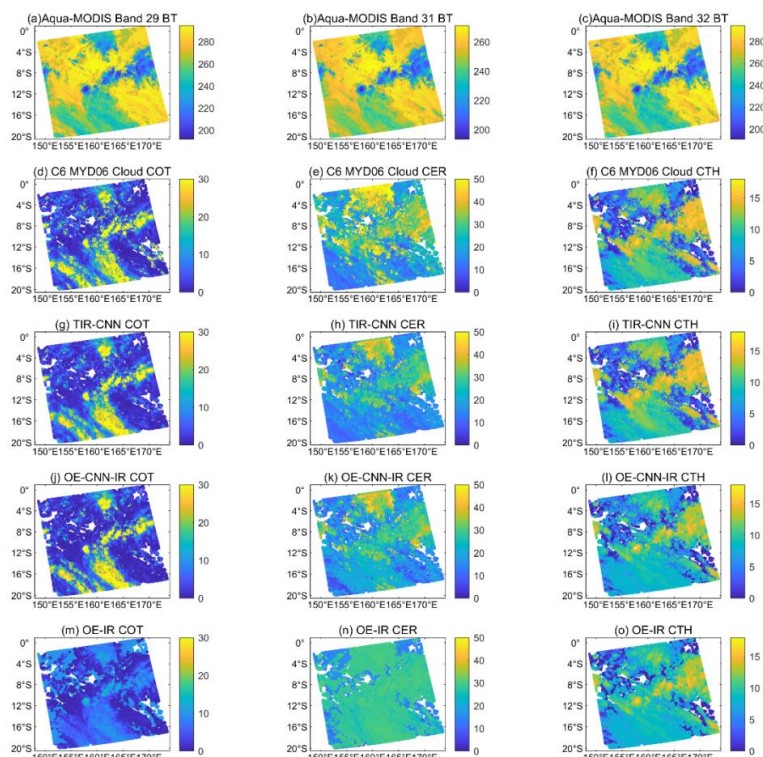


**Figure 6. Comparison of cloud properties obtained from the OE-CNN-IR model, OE-IR model and standard MODIS products for an illustrative daytime granule on 10 June. 2009 (03:00 UTC). (a, b, c) are BT image of MODIS band 29,31 and 32, respectively. (d, e, f) are the COT, CER, and CTH from the MYD06 product, respectively. (g, h, i) are the COT, CER, and CTH from the CNN-IR model, respectively. (j, k, l) are the COT, CER, and CTH from the OE-CNN-IR model, respectively. (m, n, o) are the COT, CER, and CTH from the OE-IR model, respectively.**

Fig. 7 illustrates a nighttime case of cloud parameter retrievals using CNN-IR, OE-CNN-IR and OE-IR methods. The data for this analysis is sourced from a randomly selected granule captured on February 10th, 2009, at 21:00 UTC. Figs. 7(a-c) display the BTs at channels 29, 31, and 32, and Figs. 7(d-f) shows the COT, CER, and CTH retrieved by the TIR-CNN algorithm. The relationship between COT and CTH with BT at night is generally consistent with that during the day. Figs. 7(g-i) shows the COT, CER, and CTH retrieved by the OE-CNN-IR algorithm. The OE-CNN-IR retrievals align well with the high and low-value areas in the BT images, indicating that OE-CNN-IR effectively discerns the intricate spatial variations in cloud properties during nighttime conditions. Figs. 7(j-l) display the cloud parameters retrieved using the OE-IR method. In this analysis, the predominance of values falls below





10, which signifies a more constrained retrieval scope when contrasted with the OE-CNN-IR method. Nevertheless, the distribution of CTH derived from OE-IR closely mirrors that obtained from OE-CNN-IR, affirming its dependability for estimating the top height of clouds. Additionally, both methods exhibit comparable distributions in CER.

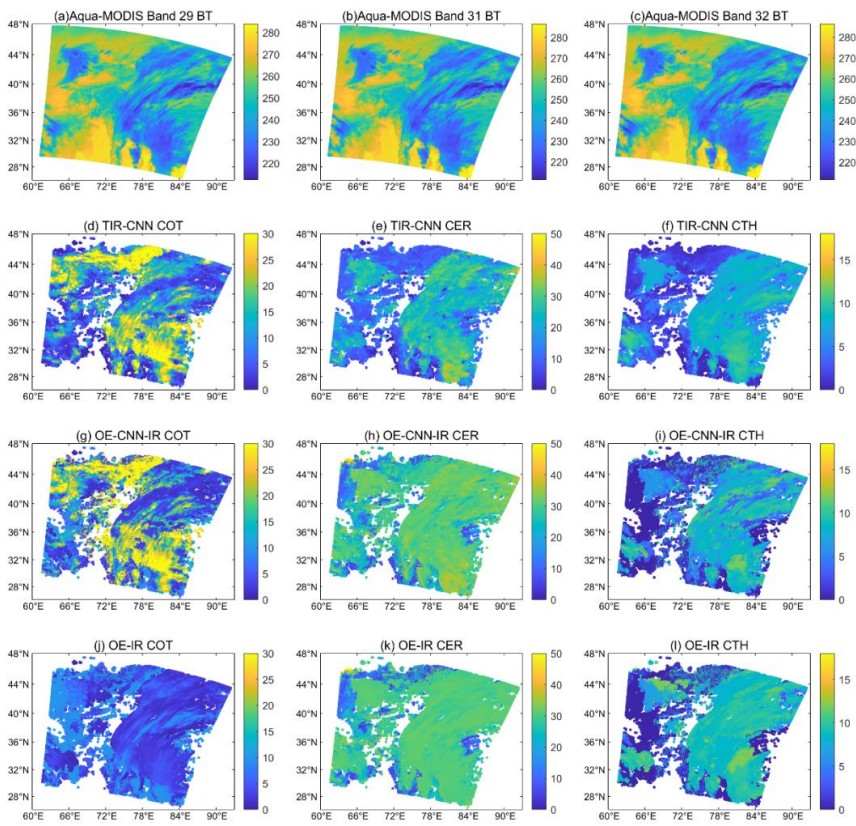


**Figure 7. Comparison of cloud properties obtained from the OE model and standard MODIS products for an illustrative nighttime granule on 10 February 2009 (21:00 UTC). (a) BT image of MODIS band 29. (b) BT image of MODIS band 31. (c) BT image of MODIS band 32. (d, e, f) are the COT, CER, and CTH from the TIR-CNN, respectively. (g, h, i) are the COT, CER, and CTH from the OE-CNN-IR model, respectively. (j,**
**k, l) are the COT, CER, and CTH from the OE-IR model, respectively.**

**3.2 Comparison between retrievals and MYD06 products in the daytime**

Figure 8 presents scatterplots that provides pixel-level comparisons of cloud property retrievals from OE-CNN-IR and OE-IR against the MYD06 ice cloud products over ocean for 2009. The left column of



Fig. 8 offers a detailed pixel-by-pixel comparison for COT, CER, and CTH between OE-CNN-IR and

the MYD06 ice cloud products. The middle column displays comparisons between MYD06 cloud

products and OE-IR retrievals. The right column displays the probability density functions obtained from

MYD06 products, OE-CNN-IR and OE-IR derived results. The color scale in these plots indicates the

number of observations in each grid, visually representing data point density. Retrieval constraints,

including a limit on Solar Zenith Angles (SZA) to less than 60 degrees and latitudes between 60°S and

60°N, ensure consistency and reliability in these comparisons. In Fig.8(a), the correlation coefficient

between OE-CNN-IR COT and MYD06 COT is 0.835 for all clouds, indicating a strong positive

correlation. In comparison, OE-IR achieves a COT correlation coefficient of 0.667 against MODIS

products for all clouds, indicating a slightly weaker relationship than that reported in Wang et al. (2016).

In Fig.8(c), the distributions provided by MYD06, OE-CNN-IR, and OE-IR are relatively similar for

COT values less than 10. The OE-CNN-IR retrievals contains a lot of cases with COT>15, which is

consistent with MODIS, but OE-IR retrievals do not contain clouds with COT>15. The underestimation

of COT for thick clouds by OE-IR is consistent to Wang et al. 2016. Therefore, it is concluded that both

OE-CNN-IR and OE-IR show consistent performance for COT below 10, but OE-CNN-IR performs

much better for thicker clouds. With respect to CER, both algorithms demonstrate moderate to weak

correlation coefficients, reflecting the inherent physical constraints of the retrieval process. Nonetheless,

OE-CNN-IR outperforms OE-IR with a correlation coefficient of 0.794, suggesting enhanced

performance. In Fig.8(f), the results from OE-IR appear to be concentrated around the priori value of 30

μm, whereas the results from OE-CNN-IR maintain a distribution that is more similar to that of MYD06.

For CTH retrieval, both OE-CNN-IR and OE-IR demonstrate good performance, with correlation

coefficients of 0.871 and 0.808, respectively. Overall, the statistical analysis in Fig. 8 underscores the

retrieval capability of OE-CNN-IR, particularly for COT and CER, compared to stand-alone OE-IR.

Figure 9 expands the ice cloud analysis from Fig. 8 to encompass all types of clouds over both land

and ocean, offering a more comprehensive evaluation of the retrieval algorithms across varied cloud

conditions. In the case of thick water clouds, the BT is not sensitive to COT, leading to most OE-IR COT

retrievals clustering around the priori value of 10. This indicates difficulties in effectively retrieving COT

for water clouds, so the OE-IR method has been used to retrieve cloud properties of ice clouds only. In

contrast, the performance of OE-CNN-IR is much better. This enhanced capability is credited to OE-



CNN-IR's effective estimation of priori states, allowing for accurate retrievals even in situation of lower BT sensitivity, as observed in water clouds. Regarding CER, the gradient of CER with respect to BT of

water clouds tends toward zero. These artifacts signal the limitations of the retrieval algorithm under minimal BT gradient conditions. Despite these challenges for CER, both OE-CNN-IR and OE-IR perform exceptionally well in retrieving CTH, with r of 0.913 and 0.931, respectively. These high correlations reflect the algorithms' effectiveness in estimating CTH.

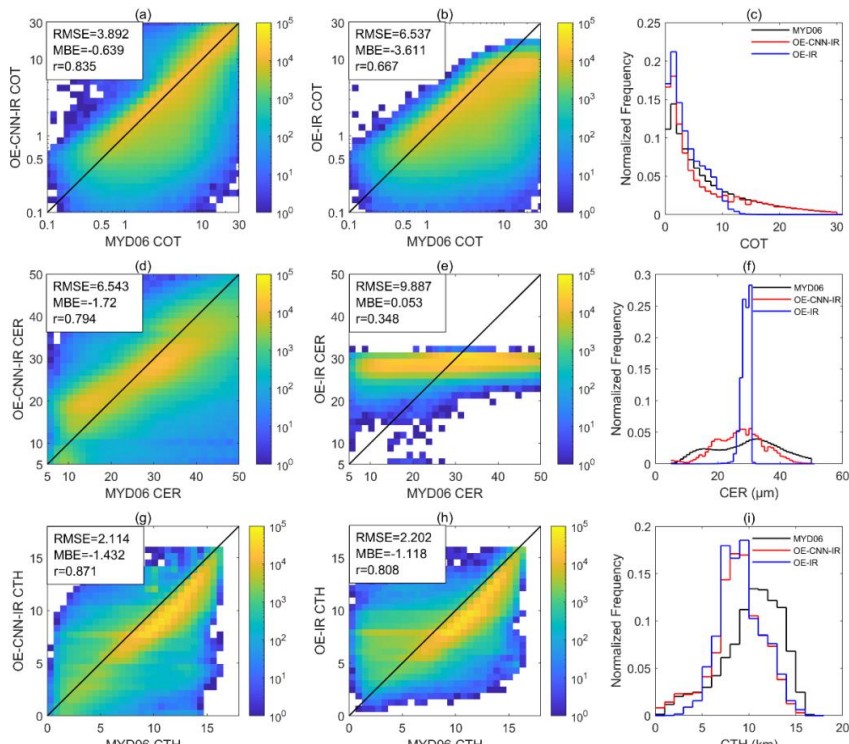

**Figure 8. Scatterplots of the pixel level comparisons between the retrievals and MYD06 products for ice clouds over oceans. (left column) Pixel-by-pixel comparisons of COT, CER, and CTH from OE-CNN-IR with the MYD06 ice cloud products over ocean in 2009. (middle column) Scatterplots of the pixel level comparisons between the MYD06 cloud products and OE-IR comparable retrievals. (right column) The probability density functions obtained from MYD06 products, OE-CNN-IR and OE-IR derived results are presented. Color**
**shadings denote the number of observations in each respective pixel. All comparable retrievals are constrained to cases with SZA < 60° and latitude between 60°S and 60°N.**

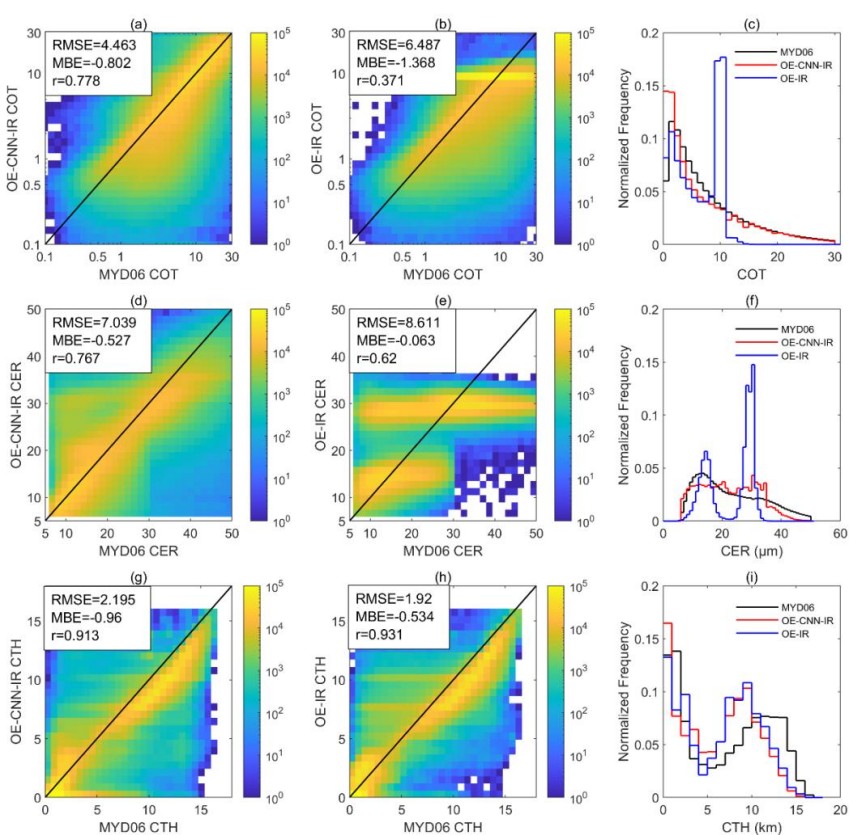

**Figure 9. Same as Fig. 8, but includes liquid clouds over ocean, and all clouds over land.**

### 3.3 Comparison with products from active sensors

Under nighttime conditions, where standard MYD06 cloud products do not offer cloud optical properties,

the evaluation is supplemented by incorporating near-real-time data from active sensors (DARDAR,

derived from CloudSat/CALIPSO observations). To align with MODIS observations, single-layer

measurements from CALIPSO are spatially matched by restricting their distance to less than 333 meters,

the distance between adjacent CALIPSO footprints. Additionally, the temporal difference is restricted to

under 90 seconds. Spatially and temporally co-located samples from 2009 are employed to evaluate the

performance and generalization capabilities of the OE model during night conditions. These criteria are

applied to achieve the closest possible data correspondence between the two different instruments,

facilitating a meaningful assessment of the OE model's nighttime performance.

Figure 10 presents a detailed comparison of COT retrievals for ice clouds using OE-CNN-IR and

OE-IR methods, benchmarked against DARDAR cloud products. The comparisons are confined to

latitudes between 60°N and 60°S to ensure a comprehensive assessment across both daytime and

nighttime conditions. The daytime correlation coefficient for OE-CNN-IR versus DARDAR COT is

0.651, with slightly lower nighttime correlation of 0.583. These values suggest a moderate concordance

of OE-CNN-IR with DARDAR, notably in the context of the challenges involved in accurately retrieving

COT for ice clouds. In contrast, OE-IR exhibits lower correlation coefficients, with 0.546 during the day

and 0.503 at night. Nevertheless, the RMSE of OE-IR is lower than that of OE-CNN-IR. Notably, the

OE-CNN-IR method demonstrate better performance for COT > 10. This suggests that OE-CNN-IR is

more adept at capturing the variability of thicker ice clouds, which is important for understanding cloud

radiative effects and their implications for weather and climate systems.

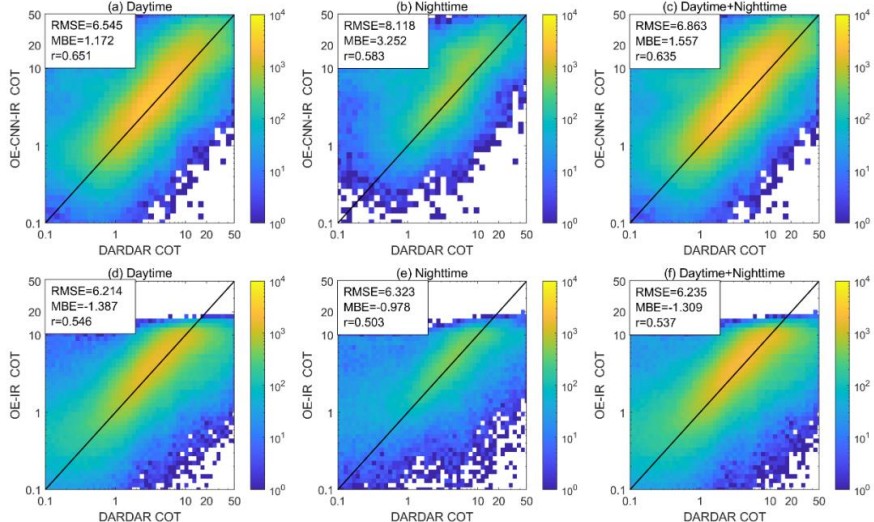

**Figure 10. Comparisons of OE-CNN-IR COT, OE-IR COT and DARDAR COT for ice clouds over oceans.**
**(a-c) are comparisons between OE-CNN-IR COT and DARDAR products. (d-f) are comparisons between**
**OE-IR COT and DARDAR products. The left column is for daytime comparisons, the middle column is for**
**nighttime comparisons, and the right column is for all-day comparisons.**

**4    Conclusions**

This study introduces a cloud property retrieval method based on optimal estimation (OE-CNN-IR),

which integrates traditional radiative transfer simulations with a machine-learning method. Designed for

retrieving COT, CER, and CTH, this method is applicable for passive satellite imagery under both



daytime and nighttime conditions. Retrievals from a machine learning algorithm (TIR-CNN) are used to

provide priori values for OE iteration, and an RTM is used to create radiance lookup tables that are used

in the iteration processes. Subsequently, the retrievals are iteratively adjusted to minimize discrepancies

between the IR observations and radiative transfer model simulations. The efficacy of OE-CNN-IR is

validated against MYD06 products and active sensor cloud products, and the results are compared to a

stand-alone optimal estimation model (OE-IR).

The validation results reveal that the OE-CNN-IR method outperforms stand-alone OE-IR model,

especially for cloud optical thickness of thick clouds. Correlation coefficients with MYD06 products

have exhibit marked improvements: correlation coefficients for COT increases from 0.667 to 0.835,

correlation coefficients for CER increases from 0.348 to 0.794, and correlation coefficients for CTH

increases from 0.808 to 0.871. In nighttime evaluations, the OE-CNN-IR method consistently

outperforms the traditional OE model when compared with DARDAR COT. The consistency between

OE-CNN-IR retrievals and MYD06 products is as good as that of stand-alone machine-learning retrieval

algorithm (i. e., TIR-CNN), and the radiance simulations based on OE-CNN-IR retrievals exhibit greater

consistency with actual observations, as depicted in Fig. 5. Furthermore, the algorithm explicitly

addresses physical processes, aligning with the preferences of scientists who advocate for physically

based methodologies. While the OE-CNN-IR method in this study is primarily applied to Aqua-MODIS

imagery, it can be potentially applied to other sensors with similar infrared (IR) channels. For instance,

it can be readily adapted to geostationary satellites, given their analogous wavelength ranges(Tana et al.,

2023; Zhao et al. 2023). In the future, the combination of machine-learning algorithms and traditional

radiative transfer simulations might be further developed in other fields of remote sensing.

**Code/Data availability**

The custom code/data used in this study is available upon reasonable request from the corresponding

author and related OE codes are available at https://github.com/huanghe141327/Codes-for-Huang-et-al..

**Competing interests**

The authors declare that they have no conflict of interest.



**Credit authorship contribution statement**

**He Huang:** Methodology, Data curation, Validation, Formal analysis. **Quan Wang:** Data curation, Writing – review & editing. **Chao Liu:** Writing – review & editing. **Chen Zhou:** Conceptualization, Methodology, Validation, Writing – review & editing, Supervision.

**Acknowledgement**

This work was supported by the National Natural Science Foundation of China (Grant No. NSFC 42075127 and 42375038), and the AI & AI for Science Project of Nanjing University.

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
