# Peer review of "Optimal estimation of cloud properties from thermal infrared observations with a combination of deep learning and radiative transfer simulation"

_Atmospheric Measurement Techniques, 2024_

## Referee Comment (RC1)

**Comments on "Optimal estimation of cloud properties from thermal infrared observations with a combination of deep learning and radiative transfer simulation" by He Huang et al. (https://doi.org/10.5194/amt-2024-87)**

The manuscript proposes a retrieval method to infer cloud optical thickness, cloud effective radius, and cloud top height from satellite thermal-infrared radiance measurements. The retrieval uses a synergetic method that combines optimal estimation (OE) and a convolutional neural network (CNN), and builds on the idea that the CNN provides the a-priori state on which the OE is applied.

 The authors validate their retrieval techniques against MODIS products (daytime observations) and DARDAR products (nighttime observations). The authors claim a good agreement between their results and the MODIS or DARDAR retrieval products. However, I am not convinced about the effectiveness of the retrieval, nor the novelty and advantage of the proposed method over the existing MODIS and DARDAR retrievals. First, this may be a consequence of the imprecise and ambiguous presentation of the results. Second, the paper has flaws in the presentation of the algorithm, the technique, and the analysis of the results. Several, easy to avoid typos make the manuscript difficult to understand. It raises the impression that the manuscript was not well prepared.

After re-reading the manuscript several times, I have came to the conclusion that I do not recommend this manuscript for publication. The manuscript should only be considered for a second round of reviews after major revisions. The new manuscript version should explicitly highlight the advantages of the retrieval method compared to existing retrievals. The presentation, from typos over grammar to structure, must be improved. More specific comments are listed below

**Major comment:**

(1) Line 107-108: The retrieval results are verified with MODIS or DARDAR products. MODIS or DARDAR retrievals themselves are based on assumptions, are maybe biased, and are subject to uncertainties. I am not questioning the quality of the MODIS or DARDAR products, but by using these products as a reference, the proposed retrieval will, in my opinion, never be better than the existing retrievals.
Have you considered using "synthetic observations" based on large-eddy simulations (LES) coupled with radiative transfer (RT) simulations? Such LES and RT simulations would provide ground truth values of COT, CER, CTH, and radiances, and would allow a real evaluation of the proposed retrieval method.

(2) No information is provided about the convolutional neural network (CNN). In particular, the type of CNN, how it is configured, what data are used for training, how many MODIS granules are used during the training, etc? For the manuscript to be useful for the community, the CNN needs more explanation. Otherwise, the retrieval appears to be the typical "black box".

(3) The individual retrieval methods should be explained more clearly to highlight their differences, advantages, and potential disadvantages. In this context, Fig. 1 should be used.

(4) Regarding the general approach. It is not clear to me why CNN and OE need to be combined or what the physical rational supports this idea. CNN and OE are two different methods that could be run independently. Why run a CNN and add an OE approach?  The authors state that CNNs are able to learn the complex non-linear relationships between observed radiance and cloud microphysical properties, and that CNN account for the spatial context. One could

question if CNNs are not as good as claimed in the introduction. Does adding OE onto a CNN retrieval really remove the criticism of CNN (Lines 81-85)?

(5) If CNN and OE are combined, would it not be better to go the other way and to improve the OE results with the CNN? This would be a well established approach, where CNN are used to improve existing retrieval methods.

(6) The analysis lacks a clear story that shows which method – OE-IR or OE-CNN-IR – is superior to the other and why.

(7) Line 262: "with those derived from CRTM" It is unclear to me what is happening here. Did you use the MODIS retrieved values of COT, CTH and CER, and put them into CRTM to make radiance simulations? Are these the values plotted on the y-axis with label "MYD06+CRTM" in Fig. 5 a,b,c ? If these are MODIS retrieval results and the resulting radiances / BT, do you have any explanation why the MODIS BT are higher than the simulations? Is there still a difference in the ice optical properties despite of the correction factor of 0.4.

**Minor comments:**

- Please check for consistent use of the serial comma.

- Please check all units. There are typos throughout the paper, e.g., use km instead of KM

- "Earth" is sometimes capitalized and sometimes not. Please be consistent.

Line 44: Instead of "signifies" do you mean "depends on"?

Line 45: You could abbreviate "cloud effective radius" as it is done before.

Line 55: Citation of Iwabuchi. Please replace uppercase letters with lowercase letters.

Line 63: The word "nears" might be replaced with "approaches".

Line 64: Please give the wavelength range for your definition of far-infrared.

Line 65: "their limited presence" is misleading. I guess you want to say that these wavelength ranges are rarely measured / satellites are not capable of measuring in these wl ranges. Please consider to rephrase this sentence.

Lines 65-67: Plane-parallel computations are typically computationally efficient and widely applied. Contrarily, three-dimensional simulations are computational expensive. Later in the manuscript you talk about the spatial awareness of CNN for cloud retrievals. You could elaborate on the topic of 3D radiative transfer simulations if they play a role in the thermal infrared.

Lines 75-76: Are you saying that CNN methods are better than traditional look-up-table approaches? I would suggest rephrasing this sentence to better convey the main message.

Line 80: What is FY4A? Please explain.

Lines 81-85: Could you please elaborate on the meaning of " lacking direct physical interpretation" as well as "offer explicit physical interpretations". What do you mean by that?

Line 86: [..] integrate […] with. It might be "integrate […] into" or "combine […] with". Please check your grammar.

Line 90 and elsewhere: Please check weather you mean "priory" or "a-priori". In OE one usually refers to "a-priori".

Line 98: Could you please specify "global data"?

Line 98: I suggest replacing "spacecraft" by "satellite"

Lines 100-102: Please check the grammar of this sentence. "For this research, […] for this study."

Lines 100-101: It would be beneficial to list the MODIS products used, explicitly stating their spatial resolution and the quantities extracted from each product. While you do this in Table 1 it is missing in Lines 100-101.

Lines 103: Change "radiations" to "radiation" ?

Lines 108-109: "All parameters are aligned to a 5-kilometer spatial resolution grid, ensuring data and variable consistency". Does this mean you have projected all parameters on a regular 5-kilometer grid. How do you deal with parameters at 1km resolution? Do you interpolate them or do you use nearest neighbor method for the selection?

Line 109: Suggestion: Replace "universality" with "applicability" ?

Line 109: What do you mean by "variable consistency"? This is contradictory, you could rephrase the sentence to be clearer.

Line 110: How do you choose a day from each month? By chance or by some specific method or criteria?

Lines 113-114: "By selecting days representative of each month, we aim to assess the algorithm's performance under varying seasonal and weather patterns." You wrote this 2 sentences before. Please check, rearrange, or delete for more concise writing

Table 1: Could you provide the wavelength ranges for the bands? The unit of cloud water path should be "$kg/m^2$", using a lower case "k". Spectral bands itself do not have a unit. They depict a wavelength range. What you mean is radiance instead of Band X? Please correct.

Line124: replace "adept" with "suited to"?

Line 131: Before writing about "our retrieval methods" could you please introduce the models that you are using? You could use Fig.1 to better introduce your general retrieval concept. Fig. 1 is only mentioned once and never explicitly explained.

Line 131: "… including temperature,…." Does this mean that you incorporate all ERA5 variables in your analysis? "including" suggests that you use all of them, not just a few. Please explicitly state which data from ERA5 you are using.

Line 131 / Fig1: After the introduction of Figure1 it would be very beneficial if the authors could explain at least one iteration of the model. Particularly highlighting the difference between the OE-IR and the OE-CNN-IR method.

Line 133: Do you construct look-up-tables based on the ERA5 data, i.e., a hypercube of ERA5 data, or do you construct look-up-tables of simulated radiance based on the ERA5 data?

Line 134 or earlier: When introducing ERA5 please cite Hersbach et al. 2020. The full citation is given at the end of the report.

Line 138: "This method iteratively adjusts parameters to reconcile observed data with model predictions." Isn't it the other way around? As I understand OE, the inputs to the forward model are iteratively adjusted such that the model output closely matches the observations. Please check.

Line 148-149: " and is able to simulate the radiances observed by satellites. " This has already been mentioned above. Please rephrase the paragraph and remove duplicate text.

Line 150: Suggestion: Replace "corroborated" with "validated"

Lines 150-154: Please check grammar. Please also specify which habit (mixture) you are using in your forward simulations.

Line 155: "various COT,.." do you mean "combinations of". Please check and change if needed.

Table2: This table is difficult to understand because the column "Notes" mixes various information, i.e., products and parameter steps. You could also choose a better way to present the COT, CER, and CTH ranges, e.g., by giving intervals and step sizes. Please add spaces between the number and the unit (and check for this in the entire manuscript).
Please revise the table and make it more clear. Otherwise I suggest to remove the table.

Line 162: "observations in" to "observations of"

Lines 165ff: Please use sub figure labels [(a), (b),...] to guide the reader through the Figure. In that way the reader is directed to the correct sub panel, which facilitates the understanding of the text.

Line 173: In several instances the authors switch between "liquid clouds" and "water clouds". Please revise the entire text to be consistent. All clouds consist of water. To be precise one should distinguish between "ice water" and "liquid water".

Figures in general: Captions are not in bold. Only the figure number and the sub panel label, e.g., (a), is in bold.

Fig2: You changed the y-range of Fig2b but kept it constant everywhere else. It would be beneficial to keep the same range (220-280 K) for all panels. This helps to compare with Fig.3

Line 193: "standard MYD06 products" Please explicitly state which variables you are using. Or do you use all variables provided in the product?

Line 196: What is meant with "convolutions"? Please explain.

Lines 209-210: "model predictions" is ambiguous. Do you mean simulation results from the forward model?

Line 212: Suggestion: Replace "proficiency" with "capability" or "ability".

Line 217: Equation 2: Typo. Xa should be $X_a$, I guess?

Line 221: Suggestion: Replace "encapsulated" by "implemented". And it is more a gradient decent method instead of a "gradient decent iterative process". You might rephrase the sentence.

Line 226: " and is set to be the same for all three variables" Do you refer to the learning rate. If so please rephrase the text.

Line 242: If the results of the OE-CNN-IR remain at the a priori values, does this not mean that the iteration fails? It means that the information content of the measurements is not taken into account.?

Line 244: Please elaborate what you mean with "function effectively."

Line 244: How do you know that the COT is correctly determined? You minimize the cost function between the observed and simulated radiances but this does not necessarily mean that the retrieved COT is correct. If you want to prove a correct COT retrieval you should plot the true / expected COT, CER, and CTH values together with your retrieval results in Fig4.

Fig4. Please add the COT values associated with "A thin cloud case" and " A thick cloud case". What do you mean by "thin" and "thick"? Optically or geometrically. Please specify.

Line 249: A feature of OE is the estimation of the errors that are associated with the a priori and a posteriori information, i.e., the uncertainties introduced by the measurements. In my opinion, this should be included, at least mentioned, in the section of OE. If you do not plan to use the OE uncertainty estimates from the OE technique please state why.

Line250: ".. forecast errors.." do you mean errors in the forward simulation. You are not making a forecast in the traditional sense.

Line 259: It is unclear which method you are referring to: OE-CNN-IR or OE-IR.

Line 261ff: Is there a particular reason, why you chose this particular MODIS granule?

Line 261: It would be more convenient to start with the figure that is currently Fig.6 and then showing Fig5. In this way the order would be more logical by showing the spatial distribution first and then the correlation.

Line 262. You are jumping between figures. Starting with 6, going to 5 and going back to 6 again. Please rearrange the entire section 3.1 and the figures.

Lines 273-275: Do you want to say that OE-IR is better than OE-CNN-IR? This would contradict your argumentation that the combination of CNN and OE is beneficial. Please check.

Lines 292-294: This is a trivial statement. Radiances or converted brightness temperatures are used for cloud property retrieval. This is the basis of all cloud property retrievals, e.g., of the MODIS retrieval that you use as a reference.

Lines 299-301: Please explain where the substantial differences come from.

Line 202: "below a certain threshold". Please explicitly mention the threshold and give a number.

Figure6: The figure is difficult to read. Images and labels are too small, please enlarge.

Line 343: Please explain how selecting data between 60N and 60 ensures "consistency and reliability in these comparisons".

Line 346: what do you mean with "all clouds": ice and liquid water clouds? The caption of Fig 8 says ice clouds but over land and ocean. Please be more specific and rephrase.

Line 357: " to be concentrated around the a priori value of 30 µm," does this not mean that the OE retrieval is too much constrained by the a priori value and does not take the measurements into account. This would be an indication that the retrieval does not work for this value.

Line 362: "all types of clouds". Please be more specific: ice, liquid water over land or ocean?

Line 368: "effective estimation of priori states" Does this not mean that you are getting the retrieved values from ERA5? Then why use satellite data?

Line 390: How many samples do you get from 2009? Please specify.

Line 393: Please state what you mean with ""meaningful assessment".

Lines 399-400: "notably in the context of the challenges involved in accurately retrieving COT for ice clouds." This contradicts your statement in Lines 365-366 where you argue that OE-CNN-IR is well suited to retrieve optically thin ice clouds.

Line 364: What does "thick water clouds" mean? Do you mean optically thick or geometrically thick?

References
Hersbach, H., Bell, B., Berrisford, P., Hirahara, S., Horányi, A., Muñoz Sabater, J., Nicolas, J., Peubey, C., Radu, R., Schepers, D., Simmons, A., Soci, C., Abdalla, S., Abellan, X., Balsamo, G., Bechtold, P., Biavati, G., Bidlot, J., Bonavita, M., De Chiara, G., Dahlgren, P., Dee, D., Diamantakis, M., Dragani, R., Flemming, J., Forbes, R., Fuentes, M., Geer, A., Haimberger, L., Healy, S., Hogan, R. J., Hólm, E., Janisková, M., Keeley, S., Laloyaux, P., Lopez, P., Lupu, C., Radnoti, G., de Rosnay, P., Rozum, I., Vamborg, F., Villaume, S., and Thépaut, J.-N.: The ERA5 global reanalysis, Q. J. Roy. Meteor. Soc., 146, 1999–2049, https://doi.org/10.1002/qj.3803, 2020.

---

## Editor Comment (EC1)

Comments on "Optimal estimation of cloud properties from thermal infrared observations with a combination of deep learning and radiative transfer simulation" by Huang et al. (AMT-2024-87) This manuscript introduces a cloud property retrieval method "OE-CNN-IR" by integrating the optimal estimation and machine-learning methods to effectively derive the COT, CER and CTH from passive satellite imagery. The method is suitable for both daytime and noghttime conditions. Validation results reveal that the OE-CNN-IR method outperforms stand-alone OE-IR method, especially for optically thick ice clouds. The topic is within the scope of Atmospheric Measurement Techniques. However, the results and discussions in the manuscript lacks rigor, especially for the evaluation and clarification on OE-CNN-IR and TIR-CNN. Specific comments are as follow.

Line 110: "one representative day each month" what I am concerned is that how the authors choose the representative day? And all the statistical evaluation in the manuscript was based on the 12-days data? That maybe not enough and unrepresentative.

Figure 4: from this figure, the author claimed that the performance of OE-CNN-IR is better than OE-IR, but at the same time, the difference is relatively small for most results with iterations of 0 (i.e., the priori from TIR-CNN) and iterations of 100 or more (i.e., the optimal estimation from OE-CNN-IR). So how to access the optimization or necessity of the new OE-CNN-IR algorithm, or whether the alone TIR-CNN algorithm is considered to be sufficient? Since figure 6 also reveals that the COT\CER\CTH derived form TIR-CNN appear to be closer to MODIS products.

Figure 5: the author's illustration and results reflected from this figure are confusing. They claimed that the retrievals of OE-CNN-IR method align more closely with observations than TIR-CNN, which can be attributed to the OE iterations. However, the performance of OE-IR method is better than that of OE-CNN-IR method both in terms of RMSE and correlation coefficient. From my opinion, the comprehensive discussion combining radiation consistency with optical property evaluation (Figure 6) is more suitable.

Line 308-309: the performance of CER retrievals using the OE-IR method maybe not comparable to that of the OE-CNN-IR method. Please check.

Line 317: "using CNN-IR, OE-CNN-IR and OE-IR" change to "using TIR-CNN, OE-CNN-IR and OE-IR".

For the retrieval method, it is unclear that the authors used all the nine IR bands (band 27 - 36) for cloud retrieval or only the three IR bands (band 29,31,32) discussed in section 3?

Figures 8/9: Compared to the difference between OE-CNN-IR and OE-IR, what I am interested in is the difference between TIR-CNN and OE-CNN-IR, as TIR-CNN retrievals seem to be closer to MYD06 from Figures 6 and 7.

There is no discussion of the cost function throughout the manuscript, whether all inversion can achieve successful convergence?

---

## Author Comment (AC1)

**Response to Reviewer # 1**

We thank the reviewer for the valuable comments. The manuscript has been modified according to the suggestions. Below are our specific responses to the reviewer's comments.

RC=Reviewer Comments
AR=Author response
TC=Text Changes

General comments:

The manuscript proposes a retrieval method to infer cloud optical thickness, cloud effective radius, and cloud top height from satellite thermal-infrared radiance measurements. The retrieval uses a synergetic method that combines optimal estimation (OE) and a convolutional neural network (CNN), and builds on the idea that the CNN provides the a-priori state on which the OE is applied. The authors validate their retrieval techniques against MODIS products (daytime observations) and DARDAR products (nighttime observations). The authors claim a good agreement between their results and the MODIS or DARDAR retrieval products. However, I am not convinced about the effectiveness of the retrieval, nor the novelty and advantage of the proposed method over the existing MODIS and DARDAR retrievals. First, this may be a consequence of the imprecise and ambiguous presentation of the results. Second, the paper has flaws in the presentation of the algorithm, the technique, and the analysis of the results. Several, easy to avoid typos make the manuscript difficult to understand. It raises the impression that the manuscript was not well prepared. After re-reading the manuscript several times, I have came to the conclusion that I do not recommend this manuscript for publication. The manuscript should only be considered for a second round of reviews after major revisions. The new manuscript version should explicitly highlight the advantages of the retrieval method compared to existing retrievals. The presentation, from typos over grammar to structure, must be improved. More specific comments are listed below

**Reply**: We thank the reviewer for the valuable comments. We have improved the paper after addressing the reviewer's comments. We highlighted the advantages of the retrieval method compared to existing retrievals in the abstract: compared to stand-alone OE which is ineffective to retrieve the optical thickness of thick clouds, the cloud properties retrieved by the new algorithm show an overall better performance. Compared with stand-alone machine-learning based algorithm, the radiances simulated based on retrievals from the new method align more closely with observations, and physical radiative processes are handled explicitly in the new algorithm.

(1) Line 107-108: The retrieval results are verified with MODIS or DARDAR products. MODIS or DARDAR retrievals themselves are based on assumptions, are maybe biased, and are subject to uncertainties. I am not questioning the quality of the MODIS or DARDAR products, but by using these products as a reference, the proposed retrieval will, in my opinion, never be better than the existing retrievals. Have you considered using "synthetic observations" based on large-eddy simulations (LES) coupled with radiative transfer (RT) simulations? Such LES and RT simulations would provide ground truth values of COT, CER, CTH, and radiances, and would allow a real evaluation of the proposed retrieval method.

**Reply:** Thanks for the comments. It is true that all observations have uncertainties. As you

mentioned, both MODIS and DARDAR have been demonstrated to be stable and accurate in climate research therefore, so they are used to test this method. Therefore, we first conducted training and inversion using MODIS daytime data, and then validated the training accuracy targeted at MODIS using DARDAR nighttime data to ensure the stability of this algorithm.

The advantage of OE-CNN-IR method can be demonstrated by Fig. 8c: the retrievals of stand-alone OE-IR method fails to retrieve COT greater than 15 (COT of deep convective clouds is typically greater than this value), but OE-CNN-IR successfully captures these thick clouds, so clearly OE-CNN-IR method is better than stand-alone OE-IR in capturing thick clouds.

Thanks for the suggestions on LES coupled with RT simulations, which would be a good tool if LES and RT are accurate enough. However, uncertainties induced by parameterizations exist in in both LES and RT. In this work, the spatial statistic relationship between adjacent cloud pixels is used by OE-CNN-IR, but the clouds simulated by LES might be different from realistic clouds, so additional retrieval uncertainties would be induced by LES simulations. It is difficult to evaluate the uncertainties induced by LES, so we keep using MODIS+DARDAR validations.

**(2) No information is provided about the convolutional neural network (CNN). In particular, the type of CNN, how it is configured, what data are used for training, how many MODIS granules are used during the training, etc? For the manuscript to be useful for the community, the CNN needs more explanation. Otherwise, the retrieval appears to be the typical "black box".**

**Reply:** Thanks for the comment. We added information on this. The CNN model architecture is shown in the following figure (Wang et. al.2022):

[Figure]

**Fig. 1.** The architecture of the TIR-CNN model. (a) The simple model architecture consists of encoder and decoder parts. (b) The operations within a basic convolutional block. (c) The operations within a down-sample block. (d) The operations within an up-sample block where ⊕ means concatenation.

We added to the revised manuscript (2.2.2) the following text (Lines 210-220):

The convolutional neural network using thermal infrared (TIR-CNN) model is trained with solar-independent variables (thermal infrared radiances, viewing zenith angles, and altitude) as inputs and uses standard MYD06 products (COT, CER and CTH in the daytime) as targets. To capture a comprehensive range of the Earth's surface and viewing geometries while accounting for seasonal variations, Wang et al. (2022) collected full-year granules from 2010 to create the training dataset. Products with a 10-day interval from 2011 were selected as the validation dataset during the training

phase. Additionally, the 10-day interval data from 2009, which is independent of both the training and validation datasets, served as the testing dataset. The granules were divided into samples sized 256 × 256 km. After preprocessing, there are 1,888,680 samples in the training dataset, 191,520 in the validation dataset, and 382,760 in the testing dataset. This TIR-CNN model is an asymmetric architecture, featuring an equal number of encoding and decoding layers arranged sequentially. The basic convolutional block consists of two 2D convolutional layers with 3 × 3 kernels. Each convolutional layer is followed by a batch normalization layer and a leaky rectified linear unit (Wang et.al.,2022). Through training, the model can capture context and learn the complex nonlinear relationship between the input variables and targets, which can be applied in the cloud property retrievals during both daytime and nighttime. The convolutions in the TIR-CNN model are beneficial in considering statistic information from neighbor fields in training. Theoretically, spatial distributions, optical and microphysical properties of clouds are all determined by the meteorological backgrounds, so cloud properties are statistically connected to their horizontal distributions. In addition, the effective radius of ice cloud particles are functions of cloud temperature. The CNN-based deep learning architecture is able to capture the statistical features among adjacent pixels of satellite observations as a solution for retrieving cloud optical and micro-physical properties (Wang et al., 2022, 2023), so it is able to provide more information than traditional algorithms that retrieve cloud properties from infrared radiances of individual pixels. The benefits of machine learning in IR cloud retrievals have also been demonstrated independently by the results of Tana et al. (2023) and Zhao et al. (2023).

(3) The individual retrieval methods should be explained more clearly to highlight their differences, advantages, and potential disadvantages. In this context, Fig. 1 should be used.
**Reply:** We added to the revised manuscript (2.2) the following text (Lines 145-151).

Figure 1 illustrates the architecture of our retrieval models. Initially, atmospheric parameters including temperature, humidity and ozone from the Fifth Generation of the European Centre for Medium-Range Weather Forecasts (ECMWF) Reanalysis (ERA5) are used to construct lookup tables for each 0.25°x0.25° spatial grid box. These LUTs enumerate the BT for each channel corresponding to varying COT, CTH and CER. Subsequently, the OE method is performed to retrieve cloud properties. The OE method can get the optimal solution by accounting for all spectral information. However, the iteration may have started a long way from the solution in nonlinear problem and the cost function decrease is much slower. Start with a better first guess rather than climatology value can make the process converges much more quickly (Rodgers, 2000). The deep learning methods can achieve high accuracy, and once trained, they offer very fast prediction speeds. However, due to multiple neural networks, deep learning results often lack interpretability, leading to the perception of deep learning as a black box model.

(4) Regarding the general approach. It is not clear to me why CNN and OE need to be combined or what the physical rational supports this idea. CNN and OE are two different methods that could be run independently. Why run a CNN and add an OE approach? The authors state that CNNs are able to learn the complex non-linear relationships between observed radiance and cloud microphysical properties, and that CNN account for the spatial context. One could question if CNNs are not as good as claimed in the introduction.

**Reply**: The reason that we combine CNN and OE is that both stand-alone CNN and stand-alone OE have limitations: stand-alone OE (OE-IR) systematically underestimate the COT of thick clouds, so it is not accurate enough; stand alone CNN (TIR-CNN) is more accurate, but it is frequently criticized by the lack of physical interpretability (i. e., a black-box). The combined model (OE-CNN-IR) is better than OE-IR because it could retrieve COT of thick clouds.

Does adding OE onto a CNN retrieval really remove the criticism of CNN (Lines 81-85)?
**Reply**: OE-CNN-IR is better than stand-alone CNN in two ways: 1) CNN is a pure statistic model, while OE-CNN-IR accounts for the physics, so theoretically the reliability of OE-CNN-IR is better than stand-alone CNN. 2) The radiance simulations based on OE-CNN-IR retrievals exhibit greater consistency with actual observations than stand-alone CNN. It is true that OE-CNN-IR is not perfect, but it combines the advantages of OE and CNN.

(5) If CNN and OE are combined, would it not be better to go the other way and to improve the OE results with the CNN? This would be a well established approach, where CNN are used to improve existing retrieval methods.
**Reply**: Yes, we aim to improve the OE results with CNN and the method can be used to improve existing retrieval methods as you mentioned.

(6) The analysis lacks a clear story that shows which method – OE-IR or OE-CNN-IR – is superior to the other and why.
**Reply:** OE-CNN-IR is superior than OE-IR, because the former could retrieve COT of thick clouds (COT>15) while the latter could not. In chapter Conclusions, (Line 461-462), we provided a description that 'The validation results reveal that the OE-CNN-IR method outperforms stand-alone OE-IR model, especially for cloud optical thickness of thick clouds.'

(7) Line 262: "with those derived from CRTM" It is unclear to me what is happening here. Did you use the MODIS retrieved values of COT, CTH and CER, and put them into CRTM to make radiance simulations?
**Reply**: Yes. I apologize for any confusion caused by the description. We have made the following modifications (Line 299): Figure 6 compares MODIS observed BTs with simulated BTs derived from CRTM.

Are these the values plotted on the y-axis with label "MYD06+CRTM" in Fig. 5 a,b,c ? If these are MODIS retrieval results and the resulting radiances / BT, do you have any explanation why the MODIS BT are higher than the simulations?
**Reply:** The simulated radiances depend on cloud top height (CTH) and vertical profile of air temperature. Biases in ERA5 air temperature profile and biases in MODIS CTH will lead to biases in simulated BT.

Is there still a difference in the ice optical properties despite of the correction factor of 0.4
**Reply**: Yes. The scaling factor is chosen to maximized the correlation between simulated BT and observed BT, but does not change the mean value significantly. It is likely that biases in MYD06

CTH (or ERA5 air temperature) is the primary contributor to the difference, because there is no bias when the OE-retrieved CTH is used (Fig.6 (g-l)).

Minor comments:

- Please check for consistent use of the serial comma.
**Reply:** Revised as suggested.

- Please check all units. There are typos throughout the paper, e.g., use km instead of KM- "Earth" is sometimes capitalized and sometimes not. Please be consistent.
**Reply**: Thank you for your meticulous review. 'MYD021KM' represents the product name, and we use 'km' as unit. And all 'Earth' have been changed as 'earth'.

Line 44: Instead of "signifies" do you mean "depends on"?
**Reply:** We have revised as suggested (Line 44).
This approach is grounded in the principle that cloud reflectance in non-absorbing VIS wavelengths predominantly depends on COT,

Line 45: You could abbreviate "cloud effective radius" as it is done before.
**Reply:** We have described in Line 42 'such as cloud optical thickness (COT) and cloud effective radius (CER)'.

Line 55: Citation of Iwabuchi. Please replace uppercase letters with lowercase letters.
**Reply**: Corrected in the revised manuscript.

Line 63: The word "nears" might be replaced with "approaches".
**Reply:** Corrected in the revised manuscript.

Line 64: Please give the wavelength range for your definition of far-infrared.
**Reply:** This far-infrared is the definition from (Libois et al., 2017) including band number from 1(6.8 $\mu$m) to 10 (14.2$\mu$m) in MODIS and we have added in the revised paper.

Line 65: "their limited presence" is misleading. I guess you want to say that these wavelength ranges are rarely measured / satellites are not capable of measuring in these wl ranges. Please consider to rephrase this sentence.
**Reply:** Yes. We revised the sentence as follows (Line 65):
these far infrared channels are rarely measured by satellites during the last decade.

Lines 65-67: Plane-parallel computations are typically computationally efficient and widely applied. Contrarily, three-dimensional simulations are computational expensive. Later in the manuscript you talk about the spatial awareness of CNN for cloud retrievals. You could elaborate on the topic of 3D radiative transfer simulations if they play a role in the thermal infrared.
**Reply:** Thank you for your suggestion. Here we mainly emphasize that simulating a global database consumes a lot of computing power. 3D radiative transfer simulations can be applied in our further

study.

 Are you saying that CNN methods are better than traditional look-up-table approaches? I would suggest rephrasing this sentence to better convey the main message.

**Reply:** We revised as follows (Lines 75-76):

The retrieved results show good agreements compared to both passive and active cloud products and is effective during both daytime and nighttime.

Line 80: What is FY4A? Please explain.

**Reply:** We revised as follows:

using Fengyun-4A satellite observations.

Lines 81-85: Could you please elaborate on the meaning of " lacking direct physical interpretation" as well as "offer explicit physical interpretations". What do you mean by that?

**Reply:** We mean that deep learning models, especially neural networks, consist of numerous layers and parameters (weights and biases). The interactions between these layers can be intricate, making it difficult to trace how input data is transformed into output predictions. Unlike traditional models, where parameters can be directly interpreted (like coefficients in linear regression), deep learning models do not provide straightforward explanations for their output.

The revised context is as follows (Lines 81-83):

However, the reliance of these machine learning methods on mathematical and statistical approaches typically leads to an implicit assimilation of the relationships between cloud properties and radiance observations, making it difficult to trace how input data is transformed into output predictions.

Line 86: [..] integrate […] with. It might be "integrate […] into" or "combine […] with". Please check your grammar.

**Reply:** We revised as follows:

In this study, we combine traditional radiative transfer simulations with TIR-CNN retrievals using the OE method

Line 90 and elsewhere: Please check weather you mean "priory" or "a-priori". In OE one usually refers to "a-priori".

**Reply:** Thank you for your meticulous review. We have changed all description to 'a priori'.

Line 98: Could you please specify "global data"?

**Reply:** 'Global data' refers to data that is collected, processed, and analyzed on a worldwide scale.

Line 98: I suggest replacing "spacecraft" by "satellite"

**Reply:** Revised as suggested.

Lines 100-102: Please check the grammar of this sentence. "For this research, […] for this study."

**Reply:** Thank you for your meticulous review. We revised as follows (Lines 101-103):

The Aqua-MODIS official Collection 6.1 (C6.1) products (MYD021KM, MYD03, MYD35,

MYD06 and MCD12C1), with the spatial resolution of 1 km, available at https://ladsweb.modaps.eosdis.nasa.gov/search/, have been selected for this study.

Lines 100-101: It would be beneficial to list the MODIS products used, explicitly stating their spatial resolution and the quantities extracted from each product. While you do this in Table 1 it is missing in Lines 100-101.

**Reply:** We revised as follows:

The Aqua-MODIS official Collection 6.1 (C6.1) products (MYD021KM, MYD03, MYD35, MYD06 and MCD12C1), with the spatial resolution of 1 km, available at https://ladsweb.modaps.eosdis.nasa.gov/search/, have been selected for this study.

We converted the 1 km resolution data to 5 km resolution for our analysis. The volume of data is substantial, but we haven't specifically quantified the number of data points.

Lines 103: Change "radiations" to "radiation" ?

**Reply:** Revised as suggested.

Lines 108-109: "All parameters are aligned to a 5-kilometer spatial resolution grid, ensuring data and variable consistency". Does this mean you have projected all parameters on a regular 5 kilometer grid. How do you deal with parameters at 1km resolution? Do you interpolate them or do you use nearest neighbor method for the selection?

**Reply:** We averaged the 1 km resolution data to create 5 km resolution data.

Line 109: Suggestion: Replace "universality" with "applicability" ?

**Reply:** Revised as suggested.

Line 109: What do you mean by "variable consistency"? This is contradictory, you could rephrase the sentence to be clearer.

**Reply:** We revised as follows (Line 114):

All parameters are aligned to a 5-kilometer spatial resolution grid, ensuring uniformity in data and variables.

Line 110: How do you choose a day from each month? By chance or by some specific method or criteria?

**Reply:** The default spacing between adjacent days is 30, and the spacing is set to be 40 if a date lies in the same month as previous date. We added the information to the main text (Lines 115-119):

retrievals are compared to MYD06 data from one day each month in 2009 (January 1, February 10, March 12, April 11, May 11, June 10, July 10, August 9, September 8, October 8, November 7, and December 7. The default spacing between adjacent days is 30, and the spacing is set to be 40 if a date lies in the same month as previous date)

For the comparison with DARDAR data, the whole year's data is used.

Lines 113-114: "By selecting days representative of each month, we aim to assess the algorithm's performance under varying seasonal and weather patterns." You wrote this 2 sentences before.

Please check, rearrange, or delete for more concise writing

**Reply:** We deleted this sentence.

Table 1: Could you provide the wavelength ranges for the bands? The unit of cloud water path should be "kg/m²", using a lower case "k". Spectral bands itself do not have a unit. They depict a wavelength range. What you mean is radiance instead of Band X? Please correct.

**Reply:** Revised as suggested.

Line124: replace "adept" with "suited to"?

**Reply:** Revised as suggested.

Line 131: Before writing about "our retrieval methods" could you please introduce the models that you are using? You could use Fig.1 to better introduce your general retrieval concept. Fig. 1 is only mentioned once and never explicitly explained.

**Reply:** Thank you for your suggestion and we add following description in (Lines139-140):

The core algorithm of our inversion method is the optimal estimation method, which utilizes the CRTM as the forward model and incorporates CNN results as a prior information.

Line 131: "… including temperature,…." Does this mean that you incorporate all ERA5 variables in your analysis? "including" suggests that you use all of them, not just a few. Please explicitly state which data from ERA5 you are using.

**Reply:** The revised context is as followed (Lines 141-143):

Initially, temperature, humidity and ozone from the Fifth Generation of the European Centre for Medium-Range Weather Forecasts (ECMWF) Reanalysis (ERA5) (Hersbach et al., 2020) are used to construct lookup tables for each 0.25°x0.25° spatial grid box.

Line 131 / Fig1: After the introduction of Figure1 it would be very beneficial if the authors could explain at least one iteration of the model. Particularly highlighting the difference between the OE IR and the OE-CNN-IR method.

**Reply:** We added to the revised manuscript (2.2) the following text (Lines 151-154).

In OE-CNN-IR approach, the TIR-CNN derived cloud properties provide a priori state for iterative processes, which are subsequently refined through iterative minimization of the objective cost function, while the climatology values were used as starting points in OE-IR.

Line 133: Do you construct look-up-tables based on the ERA5 data, i.e., a hypercube of ERA5 data, or do you construct look-up-tables of simulated radiance based on the ERA5 data?

**Reply:** We construct look-up-tables of simulated radiance based on the ERA5 data.

Line 134 or earlier: When introducing ERA5 please cite Hersbach et al. 2020. The full citation is given at the end of the report.

**Reply:** Thank you for your suggestion and we have cited this paper.

Line 138: "This method iteratively adjusts parameters to reconcile observed data with model

predictions." Isn't it the other way around? As I understand OE, the inputs to the forward model are iteratively adjusted such that the model output closely matches the observations. Please check.

**Reply:** We have revised as follows (Lines 154-155):

This method iteratively adjusts parameters to reconcile model predictions with observed data.

Line 148-149: " and is able to simulate the radiances observed by satellites. " This has already been mentioned above. Please rephrase the paragraph and remove duplicate text.

**Reply:** We have deleted this sentence.

Line 150: Suggestion: Replace "corroborated" with "validated"

**Reply:** Revised as suggested.

Lines 150-154: Please check grammar. Please also specify which habit (mixture) you are using in your forward simulations.

**Reply:** We use ICE_CLOUD type in CRTM and we have revised as follows (Lines 167-170):
Considering that the optical properties of ice cloud crystals in the CRTM and MODIS products differ due to variations in particle habit assumptions (Yi et al., 2016; Yao et al., 2018), the volumetric extinction cross section in the CRTM is adjusted by a scaling factor of 0.4, resulting in simulated brightness temperatures that are consistent with observations (Fig. 6(a, b, c)).

Line 155: "various COT,.." do you mean "combinations of". Please check and change if needed.

**Reply:** We revised as follows (Line 171):
For each grid cell, the CRTM simulates TIR radiances corresponding to a set of COT, CER, and CTH values at each location, from which a LUT is subsequently constructed.

Table2: This table is difficult to understand because the column "Notes" mixes various information, i.e., products and parameter steps. You could also choose a better way to present the COT, CER, and CTH ranges, e.g., by giving intervals and step sizes. Please add spaces between the number and the unit (and check for this in the entire manuscript).
Please revise the table and make it more clear. Otherwise I suggest to remove the table.

**Reply:** Revised as suggested.

Line 162: "observations in" to "observations of"

**Reply:** Revised as suggested.

Lines 165ff: Please use sub figure labels [(a), (b),...] to guide the reader through the Figure. In that way the reader is directed to the correct sub panel, which facilitates the understanding of the text.

**Reply:** Revised as suggested.

Line 173: In several instances the authors switch between "liquid clouds" and "water clouds". Please revise the entire text to be consistent. All clouds consist of water. To be precise one should distinguish between "ice water" and "liquid water".

**Reply:** We have changed all 'water cloud' to 'liquid cloud'.

Figures in general: Captions are not in bold. Only the figure number and the sub panel label, e.g., (a), is in bold.
**Reply:** Revised as suggested.

Fig2: You changed the y-range of Fig2b but kept it constant everywhere else. It would be beneficial to keep the same range (220-280 K) for all panels. This helps to compare with Fig.3
**Reply:** Thank you for your suggestion and we have revised as suggested.

Line 193: "standard MYD06 products" Please explicitly state which variables you are using. Or do you use all variables provided in the product?
**Reply:** We revised as follows (Line 210):

The convolutional neural network using thermal infrared (TIR-CNN) model is trained with solar-independent variables (thermal infrared radiances, viewing zenith angles, and altitude) as inputs and uses standard MYD06 products (COT, CER and CTH in the daytime) as targets.

Line 196: What is meant with "convolutions"? Please explain.
**Reply**: We revised as follows (Lines 218-220):

The basic convolutional block consists of two 2D convolutional layers with $3 \times 3$ kernels. Each convolutional layer is followed by a batch normalization layer and a leaky rectified linear unit (Wang et.al.,2022).

Lines 209-210: "model predictions" is ambiguous. Do you mean simulation results from the forward model?
**Reply:** Yes, and we have revised as 'model simulations'.

Line 212: Suggestion: Replace "proficiency" with "capability" or "ability".
**Reply:** Revised as suggested.

Line 217: Equation 2: Typo. Xa should be $X_a$, I guess?
**Reply:** Revised as suggested.

Line 221: Suggestion: Replace "encapsulated" by "implemented". And it is more a gradient decent method instead of a "gradient decent iterative process". You might rephrase the sentence.
**Reply:** Revised as suggested (Line 249).

Mathematically, the gradient descent method for *(i+1)*'th iteration is implemented by:

Line 226: "and is set to be the same for all three variables" Do you refer to the learning rate. If so please rephrase the text.
**Reply:** We have revised as follows (Line 254):

θ represents a learning rate and n represents the n-th cloud parameters (COT, CER and CTH), and θ is set to be the same for all three variables.

Line 242: If the results of the OE-CNN-IR remain at the a priori values, does this not mean that the

iteration fails? It means that the information content of the measurements is not taken into account.?

**Reply:** When the OE-CNN-IR retrieval is close to the priori values, it suggested that the priori values are realistic. (this is true for many cases, because the performance of stand-alone CNN model is good).

Line 244: Please elaborate what you mean with "function effectively."

**Reply:** We revised as follows (Lines 272-273):

However, for large COT values, the OE-IR method is unable to produce accurate results under these conditions.

Line 244: How do you know that the COT is correctly determined? You minimize the cost function between the observed and simulated radiances but this does not necessarily mean that the retrieved COT is correct. If you want to prove a correct COT retrieval you should plot the true / expected COT, CER, and CTH values together with your retrieval results in Fig4.

**Reply:** Revised as suggested.

Fig4. Please add the COT values associated with "A thin cloud case" and " A thick cloud case". What do you mean by "thin" and "thick"? Optically or geometrically. Please specify.

**Reply:** We denote COT<10 as thin cloud and COT>10 as thick cloud and we have added in the paper.

Line 249: A feature of OE is the estimation of the errors that are associated with the a priori and a posteriori information, i.e., the uncertainties introduced by the measurements. In my opinion, this should be included, at least mentioned, in the section of OE. If you do not plan to use the OE uncertainty estimates from the OE technique please state why.

**Reply:** Sy is the covariance matrices of the observation-to-simulation differences and it have taken these errors into account including the uncertainties introduced by the measurements.

Line250: ".. forecast errors.." do you mean errors in the forward simulation. You are not making a forecast in the traditional sense.

**Reply:** We have revised as suggested.

Line 259: It is unclear which method you are referring to: OE-CNN-IR or OE-IR.

**Reply:** We have revised in 2.2 and more detailed description have been added.

Line 261ff: Is there a particular reason, why you chose this particular MODIS granule?

**Reply:** We randomly select a granule for example.

Line 261: It would be more convenient to start with the figure that is currently Fig.6 and then showing Fig5. In this way the order would be more logical by showing the spatial distribution first and then the correlation.

**Reply:** Thanks for the suggestions. We rearranged Fig. 5 and Fig 6

Line 262. You are jumping between figures. Starting with 6, going to 5 and going back to 6 again.

Please rearrange the entire section 3.1 and the figures.

**Reply:** Thanks for the suggestions. We rearranged Fig. 5 and Fig 6, and the corresponding statements.

Lines 273-275: Do you want to say that OE-IR is better than OE-CNN-IR? This would contradict your argumentation that the combination of CNN and OE is beneficial. Please check.

**Reply:** Sorry for the misunderstanding. We have revised as follows (Lines 339-342):

The OE-CNN-IR model incorporates the OE iterations, which reduce the discrepancy between simulated and observed BT. The results indicate that retrievals of the OE-CNN-IR methods align more closely with BT observations compared to the stand-alone TIR-CNN model.

Lines 292-294: This is a trivial statement. Radiances or converted brightness temperatures are used for cloud property retrieval. This is the basis of all cloud property retrievals, e.g., of the MODIS retrieval that you use as a reference.

**Reply:** We deleted this statement.

Lines 299-301: Please explain where the substantial differences come from.

**Reply:** We added explanations on the difference (Lines 307-312):

However, significant differences are noted in CER retrieved by OE-CNN-IR and MYD06 products. This finding aligns with the work of Wang et al. (2016), which highlighted substantial discrepancies in CER retrieval when using OE-IR versus VNIR/SWIR/MWIR methods. Specifically, the C6 MYD06 cloud particle size information presented here is inferred from the 2.1 μm reflectance, which may capture signals reflected from the lower parts of a cloud (Zhang et al., 2009).

Zhang, Z., Yang, P., Kattawar, G., Riedi, J., -Labonnote, L. C., Baum, B. A., Platnick, S., and Huang, H.-L.: Influence of ice particle model on satellite ice cloud retrieval: lessons learned from MODIS and POLDER cloud product comparison, Atmos. Chem. Phys., 9, 7115–7129, doi:10.5194/acp-9-7115-2009, 2009.

Line 302: "below a certain threshold". Please explicitly mention the threshold and give a number.

**Reply:** We revised as 'below 10'.

Figure6: The figure is difficult to read. Images and labels are too small, please enlarge.

**Reply:** Revised as suggested.

Line 343: Please explain how selecting data between 60N and 60 ensures "consistency and reliability in these comparisons".

**Reply:** We have revised as follows (Lines 376-379):

Due to the large uncertainties of MODIS in retrieving COT in polar regions, retrieval constraints have been established. These include limiting the Solar Zenith Angle (SZA) to less than 60 degrees and restricting latitudes to between 60°S and 60°N, thereby ensuring consistency and reliability in these comparisons.

Line 346: what do you mean with "all clouds": ice and liquid water clouds? The caption of Fig 8

says ice clouds but over land and ocean. Please be more specific and rephrase.

**Reply:** We have revised and Fig 8 is Scatterplots of the pixel level comparisons between the retrievals and MYD06 products for ice clouds over oceans while Fig 9 is Same as Fig. 8, but includes liquid clouds over ocean, and ice and liquid clouds over land.

Line 357: "to be concentrated around the a priori value of 30 μm," does this not mean that the OE retrieval is too much constrained by the a priori value and does not take the measurements into account. This would be an indication that the retrieval does not work for this value.

**Reply:** Yes, because CER is not sensitive to thermal bands, the retrieved values are not far away from the initial values.

Line 362: "all types of clouds". Please be more specific: ice, liquid water over land or ocean?

**Reply:** We revised as follows (Line 396):

Figure 9 expands the ice cloud analysis from Fig. 8 to encompass liquid and ice clouds over both land and ocean

Line 368: "effective estimation of priori states" Does this not mean that you are getting the retrieved values from ERA5? Then why use satellite data?

**Reply**: Sorry for the misunderstanding. Cloud information from ERA5 is not used in this study. We revised the sentence as follows (Lines 400-403):

In contrast, the performance of OE-CNN-IR is much better. This shows OE-CNN-IR can be improved by using TIR-CNN outputs as a priori state, allowing for accurate retrievals even in situation of lower BT sensitivity, as observed in liquid clouds.

Line 390: How many samples do you get from 2009? Please specify.

**Reply:** There are over 4.7 million samples when compared to MYD06, and 0.54 million samples when compared to DARDAR. More detailed information has been added in 2.1.1 (Line 120):

The total sample size of MYD06 for comparison is ~4.7 million.

and in 2.1.2 (Lines 135-137):

the ice cloud product of DARDAR in 2009 is used to evaluate the inversion results during both daytime and nighttime conditions, and ~0.54 million pixels are collocated in the comparison processes.

Line 393: Please state what you mean with "meaningful assessment".

**Reply:** 'Meaningful assessment' refers to an evaluation that is both accurate and relevant, allowing for insightful conclusions about the OE model's nighttime performance. In this context, it implies that the criteria help ensure the data from the two instruments align closely enough to provide reliable insights into how well the model operates at night.

Lines 399-400: "notably in the context of the challenges involved in accurately retrieving COT for ice clouds." This contradicts your statement in Lines 365-366 where you argue that OE-CNN-IR is well suited to retrieve optically thin ice clouds.

**Reply:** We revised this sentence (Lines 431-433): The daytime correlation coefficient for OE-CNN-IR versus DARDAR COT is 0.651, with slightly lower nighttime correlation of 0.583. These values

are similar to the correlation between MYD06 COT and DARDAR COT (0.647).

Line 364: What does "thick water clouds" mean? Do you mean optically thick or geometrically thick?

**Reply:** We changed as follows (Lines 398-399):

In the case of liquid clouds above 10, the BT is not sensitive to COT, leading to most OE-IR COT retrievals clustering around value of 10.

---

## Author Comment (AC2)

Response to Reviewer # 2

We thank the reviewer for his review and valuable comments. The manuscript has been modified according to the suggestions proposed by the reviewer. The remainder is devoted to the specific response item-by-item of the reviewer's comments.

RC=Reviewer Comments
AR=Author response
TC=Text Changes

General comments:

This study proposed a cloud properties retrieval algorithm combining the optimal estimation (OE) method and convolutional neural network (CNN) method based on the infrared (IR) bands, in which the CNN-IR provides the a priori information of COT, CER, and CTH. Results indicate that the OE-CNN-IR method generally performs better than the stand-alone OE method (i.e., OE-IR) with fixed a priori values. In addition, OE-CNN-IR can retrieve all-day cloud properties that traditional two-channel methods using VIS and SWIR bands fail. The main concerns need to be addressed before accepting the manuscript.

**Reply**: We thank the reviewer for the valuable comments and suggestions. The paper has been improved after addressing all the comments.

1.In terms of methodology, the authors need to be more specific about what improvements OE-CNN-IR has and the motivation for the combination of OE and CNN, in comparison to the TIR-CNN method, as the OE-CNN-IR iterative process is highly dependent on the priori information that TIR-CNN provides. Particularly, the cloud properties derived by TIR-CNN seem to have higher consistencies with those of MYD06 than OE-CNN-IR in Fig. 6.

**Reply:** Thank you for your suggestions and we have revised 2.2 as follows (Lines 139-155):

The core algorithm of our inversion method is the optimal estimation method, which utilizes the CRTM as the forward model and incorporates CNN results as a prior information. Figure 1 illustrates the architecture of our retrieval models. Initially, temperature, humidity and ozone from the Fifth Generation of the European Centre for Medium-Range Weather Forecasts (ECMWF) Reanalysis (ERA5) (Hersbach et al., 2020) are used to construct lookup tables for each 0.25°x0.25° spatial grid box. These LUTs enumerate the BT for each channel corresponding to varying COT, CTH and CER. Subsequently, the OE method is performed to retrieve cloud properties. The OE method can get the optimal solution by accounting for all spectral information. However, the iteration may have started a long way from the solution in nonlinear problem and the cost function decrease is much slower. Start with a better first guess rather than climatology value can make the process converges much more quickly (Rodgers, 2000). The deep learning methods can achieve high accuracy, and once trained, they offer very fast prediction speeds. However, due to multiple neural networks, deep learning results often lack interpretability, leading to the perception of deep learning as a black box model. In OE-CNN-IR approach, the TIR-CNN derived cloud properties provide a priori state for iterative processes, which is subsequently refined through iterative minimization of the objective cost function, while the climatology values were used as starting

points in OE-IR. This method iteratively adjusts parameters to reconcile radiative transfer simulations with observed data. Further details are presented below.

2.P3L83, please clarify the main purpose of this study instead of 'A great number of cloud property users favor remote sensing products that offer explicit physical interpretations', which is too arbitrary. In my opinion, one of the biggest advantages of OE compared with CNN is that it could provide retrieval uncertainty, while CNN fails. In this case, any information on the retrieval uncertainty might be more valuable.

**Reply:** Thanks for the suggestion. We mean that deep learning models, especially neural networks, consist of numerous layers and parameters (weights and biases). The interactions between these layers can be intricate, making it difficult to trace how input data is transformed into output predictions. Unlike traditional models, where parameters can be directly interpreted (like coefficients in linear regression), deep learning models do not provide straightforward explanations for their outputs, and this is the primary criticize on deep learning models in discussions. It is worth noting that CNN model is able to provide an uncertainty interval on retrievals using the bootstrapping method.

3.In addition, the authors should have provided more explanation and physical meaning on the OE-CNN-IR. To emphasize the advantage of OE, I encourage the authors to extend the study of Fig. 4 using the synthetic data, by conducting information content analysis to check the best combination of available wavelengths, investigating the effects of values of Sy, Sa and Xa on the retrieval, error component analysis, etc.

**Reply:** Thank you for suggestions and more detailed description have been added in chapter 2.2 as follows (Lines 151-154):

In OE-CNN-IR approach, the TIR-CNN derived cloud properties provide a priori state for iterative processes, which is subsequently refined through iterative minimization of the objective cost function, while the climatology values were used as starting points in OE-IR.

For the chosen of channels, we added the following statement (Lines 106-108):

All channels with wavelength greater than 6.5μm are used, except that the 30th channel (primarily used for ozone retrievals) is not used to reduce uncertainties induced by ozone.

4.Table 1, the detailed wavelength information should be provided, in addition, why is the solar zenith angle excluded in the algorithm? What is the meaning of 'cloud phase infrared', 'cloud phase optical properties'?

**Reply:** The detailed wavelength information has been added. Solar zenith angle was limited less than 60° for comparison and I have added this parameter in table 1. More detailed information has been added as follows (Lines 110-114):

The product in Table 1, reported in Cloud Phase Optical Properties is the daytime-only phase used in the MYD06 cloud optical retrievals and Cloud Phase Infrared is a daytime and nighttime product derived from three IR window channel pairs. Cloud Phase Optical Properties is used in daytime to determine cloud types while Cloud Phase Infrared is used in nighttime only in our paper.

5.In Figure 1, the flowchart is relatively simple, and some details of the inversion are still unclear, e.g., what is the priori information, is there any cloud phase detection, etc.?

**Reply:** Thank you for your suggestions and more detailed description has been added in 2.2 as mentioned in General comment 1.

The sensitivity analysis in Figure 2 shows that when the COT is larger than 10, the changes of BT caused by the COT are no longer obvious, are the retrieval results reliable in the larger COT conditions?

**Reply:** When COT exceeds 10, the infrared radiances are no longer sensitive to COT, so the inversion results primarily depend on the TIR-CNN outputs, which have good agreements with the MODIS results (Wang et al. 2022), so the retrievals are still good, though less accurate than thinner clouds.

Fig.5 is a little bit confusing, since the authors want to emphasize the advantage of OE-CNN-IR, while the simulated BT based on the OE-CNN are more consistent with the observation than those of OE-CNN-IR. Then the readers might understand that OE-IR can get a more accurate retrieval through a better fitting of the observed spectral. In this case, I suggest authors provide more information on the retrieval, such as the degree of freedom.

**Reply:** Thanks for the suggestion. Here, we present a comparison between the simulated brightness temperatures from CRTM under different cloud property inputs and the observed values. The results adjusted by OE show improved performance over those obtained from the TIR-CNN. The OE-IR BT (brightness temperature) results are slightly better than those of OE-CNN-IR BT. This is primarily because OE-CNN-IR relies on the results from TIR-CNN for its iterations, and since Sa has a significant weight in the cost function, the weight of (y - F(X)) is reduced, while Sa can be ignored in OE-IR. As a result, the comparison between the converged simulated brightness temperatures and the observations is slightly less favorable.

More detailed information has been added in 2.2.3 (Lines 251-252):

When the uncertainties of a priori state are large (e. g., OE-IR), the cost function J is primarily influenced by the first term in OE-IR. If the uncertainties of a priori state are small, then the second term is also important in the iteration process (e. g., OE-CNN-IR).

P11L227, iterations over 200 times seem to be meaningless since the cost function has converged after several times iterations according to Fig.4. In addition, there is no information on the 'real' COT, CER, and CTH in Fig. 4.

**Reply:** We added MYD06 values as reference in Fig.4 as suggested.

Section 2.1.2, the title of "Active Lidar Detection cloud products" is misleading as the DARDAR product is based on CALIOP and CPR observations.

**Reply:** Thank you for pointing our issues and the title has been revised as 'Lidar-radar Detection cloud products'.

---

## Author Comment (AC3)

Response to Reviewer # 3

We thank the reviewer for the valuable comments. The manuscript has been modified according to the suggestions. The remainder is devoted to the specific response item-by-item of the reviewer's comments.

RC=Reviewer Comments
AR=Author response
TC=Text Changes

This manuscript introduces a cloud property retrieval method "OE-CNN-IR" by integrating the optimal estimation and machine-learning methods to effectively derive the COT, CER and CTH from passive satellite imagery. The method is suitable for both daytime and noghttime conditions. Validation results reveal that the OE-CNN-IR method outperforms stand-alone OE-IR method, especially for optically thick ice clouds. The topic is within the scope of Atmospheric Measurement Techniques. However, the results and discussions in the manuscript lacks rigor, especially for the evaluation and clarification on OE-CNN-IR and TIR-CNN. Specific comments are as follow.

**Reply**: We thank the reviewer for the valuable comments and suggestions. The paper has been improved after addressing all the comments.

Specific comments:
Line 110: "one representative day each month" what I am concerned is that how the authors choose the representative day? And all the statistical evaluation in the manuscript was based on the 12-days data? That maybe not enough and unrepresentative.

**Reply:** We have added more detailed information in 2.1.1 (Lines 117-122):

retrievals are compared to MYD06 data from one day each month in 2009 (January 1, February 10, March 12, April 11, May 11, June 10, July 10, August 9, September 8, October 8, November 7, and December 7. The default spacing between adjacent days is 30, and the spacing is set to be 40 if a date lies in the same month as previous date).

…The total sample size of MYD06 for comparison is ~4.7 million.

and in 2.1.2 (Lines 137-139):

the ice cloud product of DARDAR in 2009 is used to evaluate the inversion results during both daytime and nighttime conditions, and ~0.54 million pixels are collocated in the comparison processes.

(Note that only a small portion of MODIS data is collocated with DARDAR. For the comparison of MYD06, there are plenty of pixels, and the computational cost is much higher if we perform a full year of OE-IR calculations, so only 12 days are chosen. The number of pixels collocated with DARDAR is not as large, so data in the whole year is used).

Figure 4: from this figure, the author claimed that the performance of OE-CNN-IR is better than OE-IR, but at the same time, the difference is relatively small for most results with iterations of 0 (i.e., the priori from TIR-CNN) and iterations of 100 or more (i.e., the optimal estimation from OE-CNN-IR). So how to access the optimization or necessity of the new OE-CNN-IR algorithm, or whether the alone TIR-CNN algorithm is considered to be sufficient? Since figure 6 also reveals that the COT\CER\CTH derived form TIR-CNN appear to be closer to MODIS products.

**Reply:** The retrieved COT, CER and CTH based on TIR-CNN method showed good agreements with MODIS products for both daytime and nighttime in (Wang et al., 2022). Based on this, we chose the TIR-CNN results as a priori. As a result, with a better first guess, the iteration converged quickly. The new Figure 6 illustrates the role of OE in the inversion process, showing that it helps align the simulated brightness temperatures more closely with the observations.

Figure 5: the author's illustration and results reflected from this figure are confusing. They claimed that the retrievals of OE-CNN-IR method align more closely with observations than TIR-CNN, which can be attributed to the OE iterations. However, the performance of OE-IR method is better than that of OE-CNN-IR method both in terms of RMSE and correlation coefficient. From my opinion, the comprehensive discussion combining radiation consistency with optical property evaluation (Figure 6) is more suitable.

**Reply:** Thank you for suggestions and we have swapped the order of Figures 5 and 6 and rewritten Section 3.1.

The OE-IR BT (brightness temperature) results are slightly better than those of OE-CNN-IR BT. This is primarily because OE-CNN-IR relies on the results from TIR-CNN for its iterations, and since Sa has a significant weight in the cost function, the weight of (y - F(X)) is reduced, while Sa can be ignored in OE-IR. As a result, the comparison between the converged simulated brightness temperatures and the observations is slightly less favorable.

Line 308-309: the performance of CER retrievals using the OE-IR method maybe not comparable to that of the OE-CNN-IR method. Please check.

**Reply:** Thank you for thorough review. We have made the following modifications (Lines 324-326): The performance of CTH retrievals using the OE-IR method is comparable to that of the OE-CNN-IR method while the inversion of CER is not very effective due to limitations in the physical mechanisms.

Line 317: "using CNN-IR, OE-CNN-IR and OE-IR" change to "using TIR-CNN, OE-CNN-IR and OE-IR".

**Reply:** We have revised as suggested.

For the retrieval method, it is unclear that the authors used all the nine IR bands (band 27 - 36) for cloud retrieval or only the three IR bands (band 29,31,32) discussed in section 3?

**Reply:** We have revised in 3.1 as follows (Lines 330-332): We utilized the bands 27 to 29 and 31 to 36 as listed in Table 1. However, for the sake of clarity in presentation, we only display bands 29, 31, and 33 in the Figure 6.

Figures 8/9: Compared to the difference between OE-CNN-IR and OE-IR, what I am interested in is the difference between TIR-CNN and OE-CNN-IR, as TIR-CNN retrievals seem to be closer to MYD06 from Figures 6 and 7.

**Reply:** The evaluation results show that the cloud properties retrieved by the TIR-CNN are well consistent with all available MODIS day-time products (Wang et al., 2022). Although the results from TIR-CNN are more consistent with MODIS, the simulated brightness temperature results are less accurate compared with OE method in Fig 6. Based on this, we chose the TIR-CNN results as

a priori. As a result, with a better first guess, the iteration converged quickly. The new Figure 6 illustrates the role of OE in the inversion process, showing that it helps align the simulated brightness temperatures more closely with the observations.

There is no discussion of the cost function throughout the manuscript, whether all inversion can achieve successful convergence?

**Reply:** We have added real values in Fig 4 and more detailed description have been added as follows (Lines 263-266):

Figure 4 shows the iterative variations in cost function, COT, CER and CTH for both OE-IR and OE-CNN-IR under various conditions. Both algorithms show a significant decrease with an increasing number of iterations and all iterations can achieve successful convergence. However, the initial value for OE-CNN-IR is lower than OE-IR.

---

## Referee Report (RR1)

The research "Optimal estimation of cloud properties from thermal infrared observations with a combination of deep learning and radiative transfer simulation" has made significant advancements in the field of cloud property retrieval using satellite imagery. The primary finding of the study is the successful integration of traditional radiative transfer simulations with machine learning algorithm, to retrieve cloud optical thickness (COT), cloud effective radius (CER), and cloud top height (CTH) from Moderate Resolution Imaging Spectroradiometer (MODIS) data. This method, referred to as OE-CNN-IR, is effective under both daytime and nighttime conditions, addressing a long-standing limitation of previous retrieval methods. It combines the strengths of both radiative transfer model (RTM)-based cloud retrieval methods and machine learning models. RTM-based methods are physically grounded and can accurately simulate radiance at the top of the atmosphere, while machine learning models, such as TIR-CNN, can quickly and accurately process large amounts of data. By using TIR-CNN retrievals as priori states for iterative processes in the OE method, the OE-CNN-IR method is able to reconcile observed data with physical radiative processes more effectively. I thick there are a few minor issues in the manuscript.

Minor comments:

1. The descriptions of the wavelengths for each band in Table 1 and Figure 2/Figure 3 are inconsistent.
2. In Figure 5, all parameters are lacking units. It is recommended to add the units either in the caption or above the colorbar. This problem also exist in other figures.
3. I thick "Earth" should be capitalized. Please check the full text.
4. Line 111: "6.5μm" lacks space between numbers and units, please check the full text.

---

## Author Response (AR2)

Response to Reviewer's Report

We thank the reviewer for his review and valuable comments. The manuscript has been modified according to the suggestions proposed by the reviewer. The remainder is devoted to the specific response item-by-item of the reviewer's comments.

RC=Reviewer Comments
AR=Author response
TC=Text Changes

The research "Optimal estimation of cloud properties from thermal infrared observations with a combination of deep learning and radiative transfer simulation" has made significant advancements in the field of cloud property retrieval using satellite imagery. The primary finding of the study is the successful integration of traditional radiative transfer simulations with machine learning algorithm, to retrieve cloud optical thickness (COT), cloud effective radius (CER), and cloud top height (CTH) from Moderate Resolution Imaging Spectroradiometer (MODIS) data. This method, referred to as OE-CNN-IR, is effective under both daytime and nighttime conditions, addressing a long-standing limitation of previous retrieval methods. It combines the strengths of both radiative transfer model (RTM)-based cloud retrieval methods and machine learning models. RTM-based methods are physically grounded and can accurately simulate radiance at the top of the atmosphere, while machine learning models, such as TIR-CNN, can quickly and accurately process large amounts of data. By using TIR-CNN retrievals as priori states for iterative processes in the OE method, the OE-CNN-IR method is able to reconcile observed data with physical radiative processes more effectively. I think there are a few minor issues in the manuscript.

Minor comments:
1. The descriptions of the wavelengths for each band in Table 1 and Figure 2/Figure 3 are inconsistent.

**Reply:** Thank you for pointing our issues and the wavelength in the Figure 2/Figure 3 has been corrected to the appropriate value.

[Figure]

**Figure 2. Radiative transfer model simulations for ice clouds. The atmospheric profile is from the coordinates with a longitude of 175.87° E and a latitude of 60.55° N, on June 10, 2009, at 00:00 UTC. (a) TOA BTs as a function of COT, when CER and CTH is set to 20 μm and 10 km, respectively. (b) BT as a function of CER, when COT and CTH is set to 5 and 10 km, respectively. (c) BT as a function of CTH, when COT and CER is set to 5 and 20 μm, respectively.**

[Figure]

**Figure 3. Same as Fig. 2, but for liquid clouds.**

2. In Figure 5, all parameters are lacking units. It is recommended to add the units either in the caption or above the colorbar. This problem also exist in other figures.
**Reply:** Thank you for your suggestions and we have added units in the figures.

[Figure]

**Figure 5. Comparison of cloud properties obtained from the OE-CNN-IR model, OE-IR model and standard MODIS products for an illustrative daytime granule on 10 June. 2009 (03:00 UTC). (a, b, c) are BT image of MODIS band 29,31 and 32, respectively. (d, e, f) are the COT, CER, and CTH from the MYD06 product, respectively. (g, h, i) are the COT, CER, and CTH from the CNN-IR model, respectively. (j, k, l) are the COT, CER, and CTH from the OE-CNN-IR model, respectively. (m, n, o) are the COT, CER, and CTH from the OE-IR model, respectively.**

[Figure]

**Figure 7. Comparison of cloud properties obtained from the OE model and standard MODIS products for an illustrative nighttime granule on 10 February 2009 (21:00 UTC). (a) BT image of MODIS band 29. (b) BT image of MODIS band 31. (c) BT image of MODIS band 32. (d, e, f) are the COT, CER, and CTH from the TIR-CNN, respectively. (g, h, i) are the COT, CER, and CTH from the OE-CNN-IR model, respectively. (j, k, l) are the COT, CER, and CTH from the OE-IR model, respectively.**

[Figure]

**Figure 8. Scatterplots of the pixel level comparisons between the retrievals and MYD06 products for ice clouds over oceans. (left column) Pixel-by-pixel comparisons of COT, CER, and CTH from OE-CNN-IR with the MYD06 ice cloud products over ocean in 2009. (middle column) Scatterplots of the pixel level comparisons between the MYD06 cloud products and OE-IR comparable retrievals. (right column) The probability density functions obtained from MYD06 products, OE-CNN-IR and OE-IR derived results are presented. Color shadings denote the number of observations in each respective pixel. All comparable retrievals are constrained to cases with SZA < 60° and latitude between 60° S and 60° N.**

[Figure]

**Figure 9. Same as Fig. 8, but includes liquid clouds over ocean, and ice and liquid clouds over land.**

3. I thick "Earth" should be capitalized. Please check the full text.

**Reply:** Thank you for your suggestions and all 'earth' have been modified as 'Earth'.

4. Line 111:"6.5μm" lacks space between numbers and units, please check the full Text

**Reply:** Thank you for your suggestions and '6.5μm' has been revised as '6.5 μm'.